# Efficient Curvature-Aware Hypergradient Approximation for Bilevel Optimization

Youran Dong [1]  Junfeng Yang [1]  Wei Yao [2 3]  Jin Zhang [3 2 4]

## Abstract

Bilevel optimization is a powerful tool for many machine learning problems, such as hyperparameter optimization and meta-learning. Estimating hypergradients (also known as implicit gradients) is crucial for developing gradient-based methods for bilevel optimization. In this work, we propose a computationally efficient technique for incorporating curvature information into the approximation of hypergradients and present a novel algorithmic framework based on the resulting enhanced hypergradient computation. We provide convergence rate guarantees for the proposed framework in both deterministic and stochastic scenarios, particularly showing improved computational complexity over popular gradient-based methods in the deterministic setting. This improvement in complexity arises from a careful exploitation of the hypergradient structure and the inexact Newton method. In addition to the theoretical speedup, numerical experiments demonstrate the significant practical performance benefits of incorporating curvature information.

## 1. Introduction

Bilevel optimization has been widely applied to solve enormous machine learning problems, such as hyperparameter optimization (Pedregosa, 2016; Franceschi et al., 2018), meta-learning (Franceschi et al., 2018; Rajeswaran et al., 2019; Ji et al., 2020), adversarial training (Bishop et al., 2020; Wang et al., 2021; Zhang et al., 2022), reinforcement learning (Yang et al., 2019; Liu et al., 2021), and neural

---

[1]School of Mathematics, Nanjing University, Nanjing, China [2]National Center for Applied Mathematics Shenzhen, Southern University of Science and Technology, Shenzhen, China [3]Department of Mathematics, Southern University of Science and Technology, Shenzhen, China [4]Detection Institute for Advanced Technology Longhua-Shenzhen (DIATLHSZ), Shenzhen, China. Correspondence to: Jin Zhang <zhangj9@sustech.edu.cn>.

*Proceedings of the 42$^{nd}$ International Conference on Machine Learning*, Vancouver, Canada. PMLR 267, 2025. Copyright 2025 by the author(s).

architecture search (Liu et al., 2018; Liang et al., 2019).

Bilevel optimization amounts to solving an optimization problem with a constraint defined by another optimization problem. In this work, we focus on the following bilevel optimization problem:

$$\min_{x \in \mathbb{R}^m} \Phi(x) := f(x, y^*(x))$$
$$\text{s.t. } y^*(x) = \arg\min_{y \in \mathbb{R}^n} g(x, y), \quad (1)$$

where the upper- and lower-level objective functions $f$ and $g$ are real-valued functions defined on $\mathbb{R}^m \times \mathbb{R}^n$. We assume that $g$ is strongly convex *w.r.t* the lower-level variable $y$, which guarantees the uniqueness of the lower-level solution. However, solving this bilevel optimization problem remains challenging, as $y^*(x)$ can typically only be approximated through iterative schemes (Pedregosa, 2016; Grazzi et al., 2020; Dagréou et al., 2022; Chu et al., 2024).

In large-scale scenarios, gradient-based bilevel optimization methods have gained popularity due to their effectiveness (Ghadimi & Wang, 2018; Ji et al., 2021). When solving the bilevel problem (1), it is essential to estimate the hypergradient (also known as implicit gradient), which represents the gradient of $\Phi(x)$. By the implicit function theorem, the hypergradient can be expressed as:

$$\nabla\Phi(x) = \nabla_1 f(x, y^*(x)) - \nabla_{12}^2 g(x, y^*(x)) u^*(x), \quad (2)$$

where $u^*(x) := [\nabla_{22}^2 g(x, y^*(x))]^{-1} \nabla_2 f(x, y^*(x))$. Based on the structure in (2), estimating hypergradients requires solving lower-level problems and computing the Hessian inverse-vector products. Several studies have emerged to effectively address this challenge, such as those by Ghadimi & Wang (2018); Ji et al. (2021; 2022); Arbel & Mairal (2022a); Dagréou et al. (2022). However, most existing studies primarily focus on estimating the Hessian inverse-vector products, including Neumann series approximations (Chen et al., 2021; Hong et al., 2023), conjugate gradient descent (Pedregosa, 2016; Ji et al., 2021; Arbel & Mairal, 2022a), gradient descent (Arbel & Mairal, 2022a; Dagréou et al., 2022), and subspace techniques (Yang et al., 2025). In approximating $y^*(x^k)$, Pedregosa (2016) employs the L-BFGS method to solve the lower-level problem up to

a specified tolerance. Recently, two commonly used approaches have emerged. The first approach involves performing a single (stochastic) gradient descent step (Ji et al., 2022; Dagréou et al., 2022; Hong et al., 2023). The second approach entails executing multiple (stochastic) gradient descent steps (Ghadimi & Wang, 2018; Chen et al., 2021; Ji et al., 2021; 2022; Arbel & Mairal, 2022a).

From the above, it is clear that solving lower-level problems and computing Hessian inverse-vector products are typically treated as separate tasks. However, it is important to note that the Hessian $\nabla_{22}^2 g$ in Hessian inverse-vector products originates from the lower-level objective function. Hence, hypergradient approximation has an *intrinsic structure* in which solving lower-level problems and computing Hessian inverse-vector products share the same Hessian. This intuitive structure just described has been demonstrated in Ramzi et al. (2022), which introduces SHINE, a novel method that solves the lower-level problem using quasi-Newton (qN) methods and employs the associated qN matrices, along with refinement strategies, to compute Hessian inverse-vector products.

However, the theoretical analysis of SHINE is hindered by the mixing of quasi-Newton recursion schemes and complex refinement strategies. As a result, Ramzi et al. (2022) focuses on the asymptotic convergence analysis of hypergradient approximation in a deterministic setting. The convergence rate and computational complexity are lacking. Therefore, the intuitive benefit of the intrinsic hypergradient structure has not been fully realized in Ramzi et al. (2022) or in the existing literature.

### 1.1. Contributions

This paper aims to explore and exploit the benefits of leveraging the hypergradient structure. Our contributions are summarized below.

- We propose a simple Newton-based framework, NBO, for bilevel optimization that integrates solving the lower-level problems with computing Hessian inverse-vector products. This framework is built on a new curvature-aware hypergradient approximation, which utilizes the hypergradient structure and inexact Newton methods. When each subproblem in NBO is approximately solved using a single gradient descent step initialized at zero, NBO simplifies to the well-known single-loop algorithm framework presented in Dagréou et al. (2022).

- We establish the convergence rate and computational complexity for two specific instances of NBO in both deterministic and stochastic scenarios. In particular, we demonstrate improved computational complexity compared to popular gradient-based methods in the

deterministic setting.

- We conduct numerical experiments to compare the proposed algorithms with popular gradient-based methods, demonstrating the significant practical performance benefits of incorporating curvature information.

### 1.2. Notation

We refer to the optimal value of problem (1) as $\Phi^*$. The gradient of $g$ *w.r.t* the variables $x$ and $y$ are denoted by $\nabla_1 g(x,y)$ and $\nabla_2 g(x,y)$, respectively. The Jacobian matrix of $\nabla_1 g$ and the Hessian matrix of $g$ *w.r.t* $y$ are denoted by $\nabla_{12}^2 g(x,y)$ and $\nabla_{22}^2 g(x,y)$, respectively. Unless otherwise specified, the notation $\|\cdot\|$ denotes the $\ell_2$ norm for vectors and the Frobenius norm for matrices. Furthermore, the operator norm of a matrix $Z$ is denoted by $\|Z\|_{\text{op}}$.

## 2. Curvature-Aware Bilevel Optimization Framework

### 2.1. Hypergradient Approximation

The hypergradient given by (2) is intractable in practice because it requires knowing $y^*(x)$. To address this, the seminal work (Ghadimi & Wang, 2018) approximates the hypergradient by

$$\widehat{\nabla}\Phi(x,y) := \nabla_1 f(x,y) - \nabla_{12}^2 g(x,y)u^*(x,y),$$

where $y$ is an approximation of $y^*(x)$ and $u^*(x,y) := [\nabla_{22}^2 g(x,y)]^{-1}\nabla_2 f(x,y)$. Under appropriate hypotheses, Lemma 2.2 of Ghadimi & Wang (2018) provides an error bound:

$$\|\widehat{\nabla}\Phi(x,y) - \nabla\Phi(x)\| \le C\|y^*(x) - y\|, \qquad (3)$$

where $C$ is a constant. Observe that $\nabla_{22}^2 g$ in $u^*(x,y)$ is the Hessian of $g$. Hypergradient approximation has a structure in which solving lower-level problems and computing Hessian inverse-vector products share the same Hessian.

To exploit the hypergradient structure, it is natural to utilize the Hessian $\nabla_{22}^2 g$, which provides curvature information, in solving the lower-level problem. The canonical second-order optimization scheme for this purpose is Newton's method. However, Newton's method is impractical for large-scale problems; thus, we instead consider inexact Newton methods and propose a new technique for hypergradient approximation, consisting of two steps:

(i) Given $x$ and $y$, compute an inexact solution $(v,u)$ of the linear system:

$$[\nabla_{22}^2 g(x,y)](v,u) = (\nabla_2 g(x,y), \nabla_2 f(x,y)); \quad (4)$$

(ii) Compute the approximated hypergradient:

$$d_x := \nabla_1 f(x, y - v) - \nabla_{12}^2 g(x, y - v)u. \quad (5)$$

One can easily observe that $v$ is an inexact approximation of the Newton direction $v^*(x, y) := [\nabla_{22}^2 g(x, y)]^{-1} \nabla_2 g(x, y)$, just as $u$ is an inexact approximation of $u^*(x, y)$.

The motivation is twofold: (1) Denote $y^+ := y - [\nabla_{22}^2 g(x, y)]^{-1} \nabla_2 g(x, y)$ as a single Newton step. Since $\nabla_2 g(x, y^*(x)) = 0$, and assuming that $\nabla_{22}^2 g(x, \cdot)$ is $L_{g,2}$-Lipschitz continuous, by Lemma 1.2.4 in Nesterov (2018),

$$\left\| y - [\nabla_{22}^2 g(x, y)]^{-1} \nabla_2 g(x, y) - y^*(x) \right\|$$
$$\leq \frac{1}{\mu} \left\| \nabla_2 g(x, y^*) - \nabla_2 g(x, y) - [\nabla_{22}^2 g(x, y)](y^* - y) \right\|$$
$$\leq \frac{L_{g,2}}{2\mu} \| y^*(x) - y \|^2, \tag{6}$$

where we denote $y^* := y^*(x)$ to save space when there is no ambiguity. Hence, we obtain:

$$\| \widehat{\nabla} \Phi(x, y^+) - \nabla \Phi(x) \| \leq \frac{C L_{g,2}}{2\mu} \| y^*(x) - y \|^2. \tag{7}$$

This inequality shows that a single classical Newton step can accelerate the hypergradient estimation, leading to a quadratic decay. (2) Observe that $v^*(x, y)$ and $u^*(x, y)$ share the same Hessian inverse.

## 2.2. Description of the Algorithmic Framework

Building on the new hypergradient approximation, we introduce our algorithmic framework for solving the bilevel optimization problem in (1).

First, motivated by Arbel & Mairal (2022a); Dagréou et al. (2022), solving the linear system (4) at $(x^k, y^k)$ is reformulated as minimizing the following two quadratic problems, which share the same Hessian $H_k := \nabla_{22}^2 g(x^k, y^k)$:

$$\min_v \frac{1}{2} \langle \nabla_{22}^2 g(x^k, y^k) v, v \rangle - \langle \nabla_2 g(x^k, y^k), v \rangle, \tag{8}$$

$$\min_u \frac{1}{2} \langle \nabla_{22}^2 g(x^k, y^k) u, u \rangle - \langle \nabla_2 f(x^k, y^k), u \rangle. \tag{9}$$

Second, using a warm-start procedure to initialize the solver for $u$, we write $u = u^k - w$, where $u^k$ is the current iterate of $u$. Then $w$ minimizes the following quadratic function:

$$\frac{1}{2} \|w\|_{H_k}^2 - \langle \nabla_{22}^2 g(x^k, y^k) u^k - \nabla_2 f(x^k, y^k), w \rangle, \tag{10}$$

where $\|w\|_{H_k}^2 := \langle H_k w, w \rangle$. Since computing the exact minimizers may be computationally demanding, we instead seek inexact solutions. We denote these inexact solutions by $v^k$ and $w^k$ for the quadratic problems (8) and (10).

Third, we compute the approximated hypergradient and update $x$ using an inexact hypergradient descent step. The overall procedure is summarized in Algorithm 1. Here,

---

**Algorithm 1** Newton-based framework for Bilevel Optimization (NBO)

1: **Input:** Initialize $y^0, u^0, x^0$; step size $\alpha_k$.
2: **for** $k = 0, 1, \cdots, K-1$ **do**
3:     Compute inexact Newton directions $v^k, w^k$ by minimizing the quadratic functions in (8) and (10).
4:     Update $y^{k+1}$ and $u^{k+1}$:

$$y^{k+1} = y^k - v^k; \quad u^{k+1} = u^k - w^k.$$

5:     Compute the approximated hypergradient:

$$d_x^k = \nabla_1 f(x^k, y^k) - \nabla_{12}^2 g(x^k, y^k) u^k.$$

6:     Update $x^{k+1}$:

$$x^{k+1} = x^k - \alpha_k d_x^k.$$

7: **end for**

---

slightly different from (5), we use $y^k$ and $u^k$ instead of $y^{k+1}$ and $u^{k+1}$ in $d_x^k$ to facilitate potential parallel computation. We can also update these variables in an alternating manner.

Since the subproblems are strongly convex quadratic programming problems, the proposed NBO has the potential to adapt to stochastic scenarios and be implemented using various techniques from both deterministic and stochastic optimization, similar to the frameworks outlined in Arbel & Mairal (2022a); Dagréou et al. (2022). In the following, we study two specific examples.

### 2.3. Examples

**Deterministic Setting: the NBO-GD algorithm.** In the first example, we compute inexact Newton directions in NBO using gradient descent (GD) steps with a pre-defined number of iterations $T+1$. The resulting algorithm, referred to as NBO-GD, is detailed in Algorithm 2, with its subroutine $\text{GD}(x^k, y^k, u^k; T)$ outlined in Algorithm 3, where

$$d_y^k := \nabla_2 g(x^k, y^k),$$
$$d_u^k := \nabla_{22}^2 g(x^k, y^k) u^k - \nabla_2 f(x^k, y^k).$$

*Remark* 2.1. (i) When $T = 0$, that is, when each subproblem in NBO is approximately solved using a single GD step initialized at zero, NBO-GD reduces to the single-loop algorithm framework in Dagréou et al. (2022).

(ii) The subproblems in NBO can also be approximately solved using the conjugate gradient method (Nocedal & Wright, 2006), gradient descent with a Chebyshev step size (Young, 1953), or subspace techniques (Yang et al., 2025).

(iii) It is worth noting that for a fixed $k$, the subroutine $\text{GD}(x^k, y^k, u^k; T)$ does not involve gradient computations

**Algorithm 2** NBO-GD

1: **Input:** Initialize $y^0, u^0, x^0$; number of iterations $K, T$; step size $\alpha_k$.
2: **for** $k = 0, 1, \cdots, K - 1$ **do**
3:     Compute inexact Newton directions:
$$v^k, w^k = \text{GD}(x^k, y^k, u^k; T).$$
4:     Update $y^{k+1}$ and $u^{k+1}$:
$$y^{k+1} = y^k - v^k; \quad u^{k+1} = u^k - w^k.$$
5:     Compute the approximated hypergradient:
$$d_x^k = \nabla_1 f(x^k, y^k) - \nabla_{12}^2 g(x^k, y^k) u^k.$$
6:     Update $x^{k+1}$:
$$x^{k+1} = x^k - \alpha_k d_x^k.$$
7: **end for**

---

**Algorithm 3** $\text{GD}(x^k, y^k, u^k; T)$

1: Initialize $v^{-1,k} = 0$ and $w^{-1,k} = 0$; step size $\gamma_k$.
2: **for** $t = -1, 0, \cdots, T - 1$ **do**
3:     Update
$$[v^{t+1,k}, w^{t+1,k}] \tag{11}$$
$$= [I - \gamma_k \nabla_{22}^2 g(x^k, y^k)][v^{t,k}, w^{t,k}] + \gamma_k [d_y^k, d_u^k].$$
4: **end for**
5: **Return** $v^k = v^{T,k}, w^k = w^{T,k}$.

---

but relies solely on Hessian-vector product computations, sharing the same Hessian $\nabla_{22}^2 g(x^k, y^k)$.

**Stochastic Setting: the NSBO-SGD algorithm.** We consider our framework in the stochastic setting, where

$$f(x, y) = \mathbb{E}_\xi[F(x, y; \xi)], \; g(x, y) = \mathbb{E}_\zeta[G(x, y; \zeta)]. \tag{12}$$

For simplicity, we focus on basic stochastic gradient descent (SGD) algorithms. In Algorithm 4, NSBO-SGD, an adaptation of SGD to the Newton-based framework for Stochastic Bilevel Optimization (NSBO), is presented, with its subroutine $\text{SGD}(x^k, y^k, u^k; T)$ outlined in Algorithm 5. The descent directions $D_y^k, D_u^k, D_x^k$ are unbiased estimators of $d_y^k, d_u^k, d_x^k$, given by

$$D_y^k := \nabla_2 G(x^k, y^k; B_2^k), \tag{13}$$
$$D_u^k := \nabla_{22}^2 G(x^k, y^k; B_1^k) u^k - \nabla_2 F(x^k, y^k; B_3^k), \tag{14}$$
$$D_x^k := \nabla_1 F(x^k, y^k; B_3^k) - \nabla_{12}^2 G(x^k, y^k; B_4^k) u^k. \tag{15}$$

**Algorithm 4** NSBO-SGD

1: **Input:** Initialize $y^0, u^0, x^0$; number of iterations $K, T$; step size $\alpha_k$.
2: **for** $k = 0, 1, \cdots, K - 1$ **do**
3:     Compute the inexact subsampled Newton directions:
$$v^k, w^k = \text{SGD}(x^k, y^k, u^k; T).$$
4:     Update $y^{k+1}$ and $u^{k+1}$:
$$y^{k+1} = y^k - v^k; \quad u^{k+1} = u^k - w^k.$$
5:     Update $x^{k+1}$:
$$x^{k+1} = x^k - \alpha_k D_x^k,$$
    where $D_x^k$ is unbiased estimator of $d_x^k$ in (15).
6: **end for**

---

**Algorithm 5** $\text{SGD}(x^k, y^k, u^k; T)$

1: Initialize $v^{-1,k} = 0$ and $w^{-1,k} = 0$; step size $\gamma_k$.
2: Sample batches $B_1^k, B_2^k, B_3^k$, and compute $D_y^k, D_u^k$ using (13) and (14), respectively.
3: **for** $t = -1, 0, \cdots, T - 1$ **do**
4:     Sample batch $B_1^{t,k}$ and update
$$[v^{t+1,k}, w^{t+1,k}]$$
$$= [I - \gamma_k H^{t,k}][v^{t,k}, w^{t,k}] + \gamma_k [D_y^k, D_u^k], \tag{16}$$
    where $H^{t,k} := \nabla_{22}^2 G(x^k, y^k; B_1^{t,k})$.
5: **end for**
6: **Return** $v^k = v^{T,k}, w^k = w^{T,k}$.

---

*Remark* 2.2. Inspired by recent works (Dagréou et al., 2022; Chu et al., 2024), a broad class of stochastic gradient estimation techniques, such as SAGA (Defazio et al., 2014), STORM (Cutkosky & Orabona, 2019), and PAGE (Li et al., 2021), can be incorporated into the NSBO framework.

## 3. Convergence Analysis

### 3.1. Assumptions

Before presenting the theoretical results, we introduce the assumptions that will be used throughout this paper.

**Assumption 3.1.** (a) For any $x$, $g(x, \cdot)$ is strongly convex with parameter $\mu > 0$.

(b) $\nabla g$ is Lipschitz continuous with a Lipschitz constant $L_{g,1}$, and $\nabla_{22}^2 g$ and $\nabla_{12}^2 g$ are Lipschitz continuous with a Lipschitz constant $L_{g,2}$.

(c) $\nabla f$ is Lipschitz continuous with a Lipschitz constant $L_{f,1}$, and there exists a constant $L_{f,0}$ such that

$\|\nabla_2 f(x, y^*(x))\| \leq L_{f,0}$ for all $x$.

(d) There exists a constant $C_{f,0}$ such that $\|\nabla_1 f(x, y)\| \leq C_{f,0}$ for all $x$ and $y$.

Assumptions 3.1(a)-(c) are standard in the bilevel optimization literature (Ghadimi & Wang, 2018; Chen et al., 2021; Khanduri et al., 2021; Arbel & Mairal, 2022a; Dagréou et al., 2022; Ji et al., 2022; Hong et al., 2023; Chu et al., 2024). Under Assumptions 3.1(a)-(c), the hypergradient $\nabla\Phi$ is Lipschitz continuous with a constant given by:

$$L_\Phi := L_{f,1} + \frac{2L_{f,1}L_{g,1} + L_{g,2}L_{f,0}^2}{\mu} \quad (17)$$
$$+ \frac{2L_{g,1}L_{f,0}L_{g,2} + L_{g,1}^2 L_{f,1}}{\mu^2} + \frac{L_{g,2}L_{g,1}^2 L_{f,0}}{\mu^3},$$

as established in Lemma 2.2 of Ghadimi & Wang (2018). As in Ji et al. (2021; 2022), we define $L := \max\{L_{g,1}, L_{f,1}\}$ and $\kappa = L/\mu$. Then $L_\Phi = O(\kappa^3)$.

Note that Assumption 3.1(d) is not employed in the aforementioned literature but has been adopted in Ji et al. (2021); Liu et al. (2022); Kwon et al. (2023); Yang et al. (2025) for their respective purposes. In this work, this assumption is utilized to ensure that all iteration points $y^k$ remain within a predefined neighborhood of $y^*(x^k)$, provided that the initial point $y^0$ lies within a predefined neighborhood of $y^*(x^0)$. This condition aligns with the local quadratic convergence rate of Newton's method.

### 3.2. Convergence Analysis for NBO-GD

Under the above assumptions, we establish the convergence properties of NBO-GD. The detailed proof is provided in Appendix E.1. First, we define BOX 1 to represent the set of initial points satisfying $\|y^0 - y^*(x^0)\| \leq \min\{\frac{\mu}{2L_{g,2}}, \frac{1}{2\sqrt{L_1}}\}$ and $\|u^0 - u^*(x^0)\| \leq \min\{\frac{5L_1}{2L_{g,2}}, \frac{\sqrt{L_1}}{\mu}\}$, where $L_1 := L_{f,1} + L_{g,2}\frac{L_{f,0}}{\mu}$.

**Theorem 3.2.** *Under Assumption 3.1, choose an initial iterate $(y^0, u^0, x^0)$ in BOX 1. Then, for any constant step size $\gamma_k = \gamma \leq 1/L_{g,1}$, there exists a proper constant step size $\alpha_k = \alpha = \Theta(\kappa^{-3})$ and $T \geq \Theta(\kappa)$ such that NBO-GD has the following properties:*

(a) *For all integers $K \geq 1$, $\min_{0 \leq k \leq K-1} \|\nabla\Phi(x^k)\|^2 \leq \frac{2\Phi(x^0) - 2\Phi^* + 4}{\alpha K} = O(\frac{\kappa^3}{K})$. That is, NBO-GD can find an $\epsilon$-optimal solution $\bar{x}$ (i.e., $\|\nabla\Phi(\bar{x})\|^2 \leq \epsilon$) in $K = O(\kappa^3\epsilon^{-1})$ steps.*

(b) *The computational complexity of NBO-GD is: $O(\kappa^3/\epsilon)$ gradient computations and Jacobian-vector products, and $O(\kappa^4/\epsilon)$ Hessian-vector products.*

*Remark* 3.3. One can apply the gradient descent (GD) method, as described in Appendix D, to select initial points

in BOX 1. This *one-time cost* requires $O(\kappa\log\kappa)$ gradient computations and Hessian-vector products and is included in the total computational complexity.

*Remark* 3.4. If we replace GD with the conjugate gradient method (CG) in line 3 of NBO-GD for solving the subproblems (8) and (10), we obtain NBO-CG. Our analysis in Appendix E.1 shows that NBO-CG requires fewer Hessian-vector product computations than NBO-GD, specifically $O(\kappa^{3.5}\log\kappa/\epsilon)$ Hessian-verctor products.

Theorem 3.2 provides a theoretical complexity guarantee for NBO-GD. As shown in Table 1, the computational complexity of NBO-GD improves upon that of AID-BiO (Ji et al., 2021) and AmIGO (Arbel & Mairal, 2022a). Specifically, the gradient complexity of NBO-GD surpasses the state-of-the-art result by an order of $\kappa\log\kappa$.

### 3.3. Convergence Analysis for NSBO-SGD

The following assumptions are made regarding the stochastic oracles, which will be used to analyze the convergence rate and sample complexity of NSBO-SGD in Algorithm 4.

**Assumption 3.5.** There exist positive constants $\sigma_{f,1}$, $\sigma_{g,1}$, $\sigma_{g,2}$ such that

$$\mathbb{E}\left[\|\nabla F(x, y; \xi) - \nabla f(x, y)\|^2\right] \leq \sigma_{f,1}^2,$$
$$\mathbb{E}\left[\|\nabla G(x, y; \zeta) - \nabla g(x, y)\|^2\right] \leq \sigma_{g,1}^2,$$
$$\mathbb{E}\left[\|\nabla^2 G(x, y; \zeta) - \nabla^2 g(x, y)\|^2\right] \leq \sigma_{g,2}^2.$$

**Assumption 3.6.** There is a constant $r \geq 1$ such that for any iterate $(x^k, y^k)$ generated by Algorithm 4, we have

$$\mathbb{E}\left[\|y^k - y^*(x^k)\|^2\right] \leq r\left(\mathbb{E}\left[\|y^k - y^*(x^k)\|\right]\right)^2. \quad (18)$$

This assumption, which imposes bounded moments on iterates, is commonly used in the stochastic Newton literature to establish convergence results in expectation form (Bollapragada et al., 2019; Meng et al., 2020; Berahas et al., 2020). We now establish the convergence properties of NSBO-SGD as follows. The proof details can be found in Appendix E.2. Below, we define BOX 2 to represent the set of initial points satisfying $\mathbb{E}\left[\|y^0 - y^*(x^0)\|^2\right] \leq \min\{\frac{\mu^2}{20rL_{g,2}^2}, \frac{1}{4L_1}\}$ and $\mathbb{E}\left[\|u^0 - u^*(x^0)\|^2\right] \leq \min\{\frac{4L_1^2}{5rL_{g,2}^2}, \frac{L_1}{\mu^2}\}$.

**Theorem 3.7.** *Under Assumptions 3.1, 3.5 and 3.6, choose an initial iterate $(y^0, u^0, x^0)$ in BOX 2. Then, for any constant step size $\gamma_k = \gamma \leq 1/L_{g,1}$, there exists a proper constant step size $\alpha_k = \alpha = \Theta(\kappa^{-3})$ and $T \geq \Theta(\kappa)$ such that NSBO-SGD has the following properties:*

(a) *Fix $K \geq 1$. For samples with batch sizes $|B_1^{t,k}| \geq \Theta(\kappa^2)$, $|B_1^k| \geq \Theta(\kappa K + \kappa^2)$, $|B_2^k| \geq \Theta(\kappa^3 K + \kappa^4)$, $|B_3^k| \geq \Theta(\kappa^{-1}K)$, $|B_4^k| \geq \Theta(\kappa^{-1}K)$, it holds*

*Table 1.* Comparison of the computational complexities of two NBO implementations with state-of-the-art methods in the deterministic setting, including AID-BiO (Ji et al., 2021), AmIGO (Arbel & Mairal, 2022a), No-loop AID (Ji et al., 2022), and F$^2$SA (Kwon et al., 2023; Chen et al., 2023a). Note that the dependence on $\log \kappa$ is not explicitly stated in AID-BiO and AmIGO, but it can be derived from equation (30) in Ji et al. (2021) and Proposition 10 in Arbel & Mairal (2022a).

| Algorithms | Convergence Rate | Gradient Computations | Hessian-Vector Products |
|---|---|---|---|
| AmIGO-GD (Arbel & Mairal, 2022a) | $O(\kappa^3 \epsilon^{-1})$ | $O((\kappa^4 \log \kappa)\epsilon^{-1})$ | $O((\kappa^4 \log \kappa)\epsilon^{-1})$ |
| AID-BiO (Ji et al., 2021) AmIGO-CG (Arbel & Mairal, 2022a) | $O(\kappa^3 \epsilon^{-1})$ | $O((\kappa^4 \log \kappa)\epsilon^{-1})$ | $O((\kappa^{3.5} \log \kappa)\epsilon^{-1})$ |
| No-loop AID (Ji et al., 2022) | $O(\kappa^6 \epsilon^{-1})$ | $O(\kappa^6 \epsilon^{-1})$ | $O(\kappa^6 \epsilon^{-1})$ |
| F$^2$SA (Kwon et al., 2023; Chen et al., 2023a) | $O(\kappa^3 \epsilon^{-1})$ | $O(\kappa^4 \epsilon^{-1} \log(\kappa/\epsilon))$ | / |
| NBO-GD (this paper) | $O(\kappa^3 \epsilon^{-1})$ | $O(\kappa^3 \epsilon^{-1})$ | $O(\kappa^4 \epsilon^{-1})$ |
| NBO-CG (this paper) | $O(\kappa^3 \epsilon^{-1})$ | $O(\kappa^3 \epsilon^{-1})$ | $O((\kappa^{3.5} \log \kappa)\epsilon^{-1})$ |

*that* $\min_{0 \leq k \leq K-1} \mathbb{E}\left[\|\nabla \Phi(x^k)\|^2\right] = O(\frac{\kappa^3}{K})$. *That is, NSBO-SGD can find an $\epsilon$-optimal solution in $K = O(\kappa^3 \epsilon^{-1})$ steps.*

*(b) The computational complexity of NSBO-SGD is: $O(\kappa^5 \epsilon^{-2})$ gradient complexity for F, $O(\kappa^9 \epsilon^{-2})$ gradient complexity for G, $O(\kappa^5 \epsilon^{-2})$ Jacobian-vector product complexity, and $O(\kappa^7 \epsilon^{-2})$ Hessian-vector product complexity.*

Note that the number $T$ of inner-loop steps remains at a constant level. Moreover, as shown in Table 2, the computational complexity of our stochastic algorithm, NSBO-SGD, improves upon the state-of-the-art result in AmIGO by a factor of $\log \kappa$; for comparison, refer to Proposition 10 and Corollary 4 in Arbel & Mairal (2022a). Indeed, in each outer iteration, AmIGO requires $\Theta(\kappa \log \kappa)$ gradient computations with a batch size of $\Theta(\kappa^5 \epsilon^{-1})$ for each gradient, while NSBO-SGD performs a single gradient computation with a batch size of $\Theta(\kappa^6 \epsilon^{-1})$. As a result, the complexity improvement in Theorem 3.7 is not as significant as that in Theorem 3.2.

### 3.4. Proof Sketch

In this section, we present a proof sketch for the deterministic setting as an illustrative example. The proof strategy for the stochastic case follows a similar approach. The detailed proofs of Theorem 3.2, Theorem 3.7, and Remark 3.4 are provided in Appendix E.

First, we define a Lyapunov function: $V_k = f(x^k, y^*(x^k)) - \Phi^* + b_y\|y^k - y^*(x^k)\|^2 + b_u\|u^k - u^*(x^k)\|^2$, where $b_y$ and $b_u$ are constants that depend on $\kappa$. Then, following a classical Lyapunov analysis, we establish the descent of $f(x^k, y^*(x^k))$, $\|y^k - y^*(x^k)\|$, $\|u^k - u^*(x^k)\|$,

respectively. Unlike existing bilevel optimization literature, by leveraging a one-step inexact Newton method, we prove that after taking $T \geq \Theta(\kappa)$ iterations in the subroutine, $\|y^k - y^*(x^k)\|$ satisfies the descent condition:

$$\|y^{k+1} - y^*(x^{k+1})\| \leq \frac{L_{g,2}}{2\mu}\|y^*(x^k) - y^k\|^2$$
$$+ \frac{L_{g,1}}{\mu}\|x^{k+1} - x^k\| + \frac{1}{4}\|y^k - y^*(x^k)\|.$$

Next, by selecting an appropriate constant step size $\alpha$ and applying induction while leveraging the initialization strategy, we establish that $\|y^*(x^k) - y^k\| \leq \frac{\mu}{2L_{g,2}}$ for all $k$. Hence, it follows that

$$\|y^{k+1} - y^*(x^{k+1})\|$$
$$\leq \frac{1}{2}\|y^*(x^k) - y^k\| + \frac{L_{g,1}}{\mu}\|x^{k+1} - x^k\|.$$

Combining this with the descent properties of $f(x^k, y^*(x^k))$ and $\|u^k - u^*(x^k)\|$, we derive the descent property of the Lyapunov function:

$$V_{k+1} - V_k \leq -\frac{\alpha_k}{2}\|\nabla \Phi(x^k)\|^2 - A_1\|x^{k+1} - x^k\|^2$$
$$- A_2\|y^k - y^*(x^k)\|^2 - A_3\|u^k - u^*(x^k)\|^2,$$

where $A_1$, $A_2$, $A_3$ are positive constants.

Finally, by summing over iterations using a telescoping argument and estimating an upper bound for $V_0$, we derive the results presented in Theorem 3.2.

## 4. Experiments

In this section, we present experiments to evaluate the practical performance of the proposed NBO framework. Specifically, we compare our NBO-GD and NSBO-SGD methods

*Table 2.* Comparison of the sample complexities of NSBO-SGD with state-of-the-art results in the stochastic setting, including those reported in ALSET (Chen et al., 2021), SOBA (Dagréou et al., 2022; Huang et al., 2025), AmIGO (Arbel & Mairal, 2022a), FSLA (Li et al., 2022), and F$^2$SA (Kwon et al., 2023; Chen et al., 2024a). Note that $p(\kappa)$ indicates that the explicit dependence on $\kappa$ is not provided in the corresponding references. For AmIGO, the dependence on $\log \kappa$ is derived from Proposition 10 in Arbel & Mairal (2022a).

| Algorithm | Sample Complexity | Stochastic Estimators |
|---|---|---|
| ALSET (Chen et al., 2021) | $O(\kappa^9 \epsilon^{-2} \log(\kappa/\epsilon))$ | SGD |
| SOBA (Dagréou et al., 2022; Huang et al., 2025) | $O(p(\kappa)\epsilon^{-2})$ | SGD |
| AmIGO (Arbel & Mairal, 2022a) | $O((\kappa^9 \log \kappa)\epsilon^{-2})$ | SGD |
| F$^2$SA (Kwon et al., 2023; Chen et al., 2024a) | $O\left(\kappa^{11}\epsilon^{-3} \log(\kappa/\epsilon)\right)$ | SGD |
| FSLA (Li et al., 2022) | $O(p(\kappa)\epsilon^{-2})$ | SGD+$x$-momentum |
| NSBO-SGD (this paper) | $O(\kappa^9 \epsilon^{-2})$ | SGD |

with several widely used gradient-based algorithms, including SOBA, SABA (Dagréou et al., 2022), StoBiO (Ji et al., 2021), AmIGO (Arbel & Mairal, 2022a), SHINE (Ramzi et al., 2022), F2SA (Kwon et al., 2023), and MA-SABA (Chu et al., 2024). Details of the experimental setup and additional results are provided in Appendix C.

### 4.1. Synthetic Problem

First, we consider a synthetic problem to study NBO in a deterministic and controlled scenario. This problem focuses on hyperparameter optimization, where the upper-level and lower-level objective functions are defined as follows:

$$f(\lambda, \omega) = \frac{1}{|\mathcal{D}'|} \sum_{(x'_e, y'_e) \in \mathcal{D}'} \psi(\omega x'_e y'_e),$$

$$g(\lambda, \omega) = \frac{1}{|\mathcal{D}|} \sum_{(x_e, y_e) \in \mathcal{D}} \psi(\omega x_e y_e) + \frac{1}{2} \sum_{i=1}^{p} e^{\lambda_i} w_i^2,$$

where $\lambda \in \mathbb{R}^p$ represents the hyperparameter, $\omega \in \mathbb{R}^{1 \times p}$ denotes the model parameter, and $\psi(t) = \log(1 + e^{-t})$ is the logistic loss function. Here, $\mathcal{D}'$ and $\mathcal{D}$ represent the validation and training datasets, respectively. The synthetic data is generated following a procedure similar to that described in Chen et al. (2023b; 2024b); Dong et al. (2023). Specifically, the distribution of $x_e$ follows a normal distribution $\mathcal{N}(0, r'^2)$, and $y_e = w x_e + 0.1z$, where $z$ is sampled from $\mathcal{N}(0, 1)$. Subsequently, $y_e$ is transformed into binary labels: if $y_e$ exceeds the median of the dataset, it is set to 1; otherwise, it is set to $-1$.

The experimental results for $r' = 1$ are shown in Figure 1, while the results for $r' = 0.5$ and $r' = 2$ are provided in Appendix C. AmIGO is a representative method that employs multiple ($Q$) (stochastic) gradient descent steps

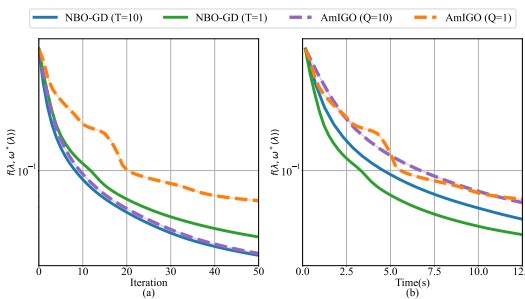

*Figure 1.* Experimental results on synthetic data with $r' = 1$.

to solve lower-level problems and quadratic subproblems related to Hessian inverse-vector products. Since we will later compare NBO-type algorithms with other gradient-based algorithms, we restrict the comparison in the synthetic problem to NBO-GD and AmIGO for simplicity.

As shown in Figure 1(a), our NBO-GD, which employs a single step of the inexact Newton method, performs comparably to AmIGO, which uses $Q = 10$ steps of gradient descent. Notably, the inner loop in NBO-GD is used to approximate the inexact Newton direction. More importantly, Figure 1(a) demonstrates that NBO-GD maintains strong performance even when the Newton direction is approximated with $T = 1$. In contrast, AmIGO's performance degrades significantly when only one step of gradient descent is used. Finally, when running time is considered, Figure 1(b) shows that NBO-GD ($T = 1$) outperforms other methods. Based on these findings, we use $T = 1$ for NBO-type algorithms in subsequent experiments.

### 4.2. Hyperparameter Optimization

In this section, we evaluate the empirical performance of our NSBO-SGD algorithm on hyperparameter selection prob-

lems, a typical bilevel optimization task (Franceschi et al., 2018; Ji et al., 2021; Dagréou et al., 2022). The upper- and lower-level objective functions are defined as follows:

$$f(\lambda, \omega) = \frac{1}{|\mathcal{D}'|} \sum_{(x'_e, y'_e) \in \mathcal{D}'} l\left(x'_e, y'_e; \omega\right),$$

$$g(\lambda, \omega) = \frac{1}{|\mathcal{D}|} \sum_{(x_e, y_e) \in \mathcal{D}} l\left(x_e, y_e; \omega\right) + r(\lambda, \omega),$$

where $\mathcal{D}'$ and $\mathcal{D}$ denote the validation and training datasets, respectively. Here, $\lambda$ is the hyperparameter, $\omega \in \mathbb{R}^{c \times p}$ denotes the model parameter, $l$ is the loss function, and $r(\lambda, \omega)$ is a regularizer.

We conduct experiments on two datasets: IJCNN1 and Covtype. The IJCNN1 dataset corresponds to a binary classification problem using logistic regression, with $p = 22$, $c = 1$. The loss function is defined as $\ell(x_e, y_e; \omega) := \psi(\omega x_e y_e)$, where $\psi$ is the logistic function. The regularizer is given by $\frac{1}{2} \sum_{i=1}^{p} e^{\lambda_i} w_i^2$. The Covtype dataset corresponds to a multi-class classification problem using logistic regression, with $p = 54$, $c = 17$. The loss function is the cross-entropy function, and the regularization term is given by $\frac{1}{2} \sum_{j=1}^{c} e^{\lambda_j} \sum_{i=1}^{p} w_{ji}^2$.

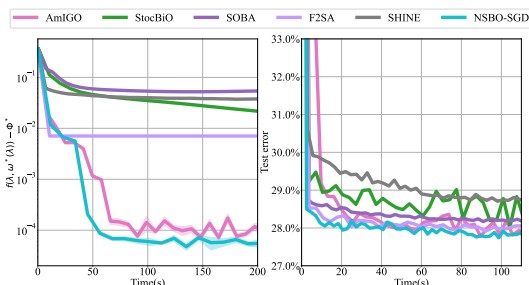

*Figure 2.* Comparison between NSBO-SGD and other algorithms on hyperparameter optimization. **Left:** IJCNN1 dataset; **Right:** Covtype dataset.

In Figure 2, we present the suboptimality gap on IJCNN1 and the test error on Covtype as functions of time. Our first observation is that, among all methods, NSBO-SGD exhibits the fastest convergence on both datasets, with its performance advantage being particularly evident on IJCNN1. Second, the gap between SOBA and AmIGO on IJCNN1 highlights the benefits of performing multiple SGD steps when solving lower-level problems, as well as handling quadratic subproblems involving Hessian inverse-vector products. In comparison, NSBO-SGD highlights the practical advantages of incorporating curvature information from the subroutine that computes an approximation of Newton directions. Notably, in Figure 2, NSBO-SGD uses $T = 1$, whereas AmIGO employs $Q = 10$. This demonstrates that the inner loop for approximating Newton directions in NSBO-SGD is more effective than the inner loop used for

solving lower-level problems and quadratic subproblems in AmIGO.

### 4.3. Data Hyper-Cleaning

We also conduct data hyper-cleaning experiments (Franceschi et al., 2017; Dagréou et al., 2022) on two datasets: MNIST and FashionMNIST (Xiao et al., 2017). Data hyper-cleaning involves training a multiclass classifier while addressing training samples with noisy labels. It can be formulated as a bilevel optimization problem with the following objective functions:

$$f(\lambda, \omega) = \frac{1}{|\mathcal{D}'|} \sum_{(x'_e, y'_e) \in \mathcal{D}'} \mathcal{L}(\omega x'_e, y'_e),$$

$$g(\lambda, \omega) = \frac{1}{|\mathcal{D}|} \sum_{(x_e, y_e) \in \mathcal{D}} \sigma(\lambda_e) \mathcal{L}(\omega x_e, y_e) + c_r \|\omega\|^2,$$

where $\mathcal{L}$ denotes the cross-entropy loss, $\sigma$ is the sigmoid function, and $c_r$ is a regularization parameter. For this experiment, we set the corruption probability to $p' = 0.5$, as in (Dagréou et al., 2022).

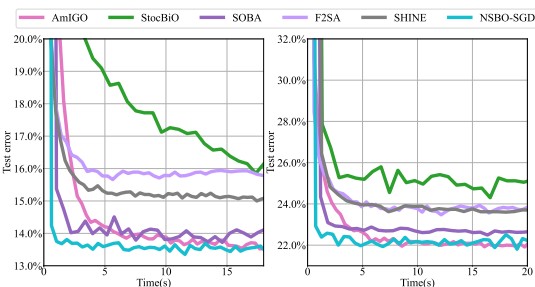

*Figure 3.* Comparison between NSBO-SGD and other algorithms on data hyper-cleaning. **Left:** MNIST dataset; **Right:** FashionMNIST dataset.

In Figure 3, we report the test errors for each method *w.r.t* running time on MNIST and FashionMNIST. We observe that both AmIGO and NSBO-SGD outperform all other methods by reaching the smallest error. Meanwhile, NSBO-SGD is the fastest, demonstrating the efficiency of incorporating curvature information in bilevel optimization.

### 4.4. Other Implementations of NBO

**Variance reduction and momentum.** Similar to the gradient-based framework in (Dagréou et al., 2022), the simplicity of our NBO approach allows us to easily incorporate variance-reduced gradient estimators and momentum techniques (Chen et al., 2024c; Chu et al., 2024). We conduct numerical experiments to test the versatility of NBO by comparing it with the framework in (Dagréou et al., 2022), using the same variance-reduced gradient estimator and momentum technique.

SABA (Dagréou et al., 2022) is an adaptation of the variance reduction algorithm SAGA (Defazio et al., 2014) for bilevel optimization. MA-SABA (Chu et al., 2024) extends the SABA algorithm by incorporating an additional standard momentum (also referred to as a moving average) into the update of the upper-level variable. We take a similar approach and extend SAGA to NBO by replacing SGD in NSBO-SGD with SAGA, resulting in a new implementation of NBO, referred to as NSBO-SAGA. If we further incorporate an additional standard momentum into the update of the upper-level variable, we obtain MA-NSBO-SAGA, another new implementation of NBO.

In Figure 4, we present the comparison results between SABA and NSBO-SAGA, as well as between MA-SABA and MA-NSBO-SAGA. We observe that both NSBO-SAGA and MA-NSBO-SAGA achieve better performance.

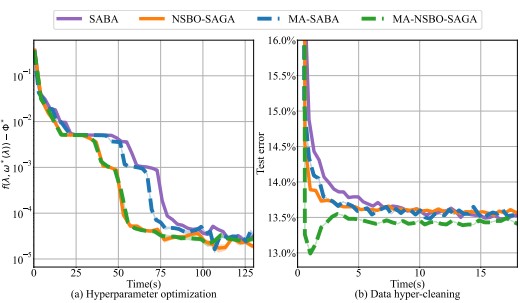

*Figure 4.* Comparison between the NBO framework and other algorithms incorporating variance reduction and the moving average technique. **Left:** Hyperparameter optimization on IJCNN1; **Right:** Data hyper-cleaning on MNIST.

**Extension to lower-level non-strongly convex structures.** The NBO framework is primarily designed for bilevel optimization problems in which the lower-level objective is strongly convex. For problems with a non-strongly convex lower-level structure, existing methods often reformulate the original problem to induce strong convexity. Once this condition is met, the NBO framework can be applied. For instance, in the BAMM method (Liu et al., 2023b), when $g$ is merely convex, an aggregation function $\phi_\mu = \mu f + (1-\mu)g$ is introduced. This function becomes strongly convex if $f$ is strongly convex. An approximate hypergradient $d_x^k$ can then be computed by substituting $g$ with $\phi_\mu$.

We compare the BAMM method and BAMM+NBO, where the NBO framework is used to compute $d_x^k$ within BAMM, on the toy example proposed in Liu et al. (2023b). The example is defined as follows:

$$\min_{x\in\mathbb{R}^n} \frac{1}{2}\|x - y_2\|^2 + \frac{1}{2}\|y_1 - \mathbf{e}\|^2$$

$$\text{s.t.} \quad y = (y_1, y_2) \in \arg\min_{(y_1,y_2)\in\mathbb{R}^{2n}} \frac{1}{2}\|y_1\|^2 - x^\top y_1,$$

where $\mathbf{e}$ is a vector with all components equal to 1. It

is straightforward to derive that the optimal solution is $(\mathbf{e}, \mathbf{e}, \mathbf{e})$. The results, shown in Figure 5, demonstrate that BAMM+NBO significantly outperforms BAMM, highlighting the effectiveness of the NBO framework.

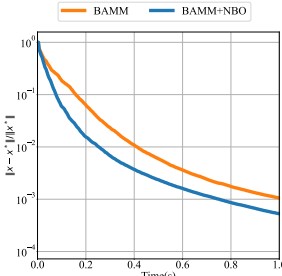

*Figure 5.* Performance of the NBO framework on a toy example with a non-strongly convex lower-level problem.

## 5. Conclusion

In this work, we design and analyze a simple and efficient framework (NBO) for bilevel optimization, leveraging the hypergradient structure and inexact Newton methods. The convergence analysis and experimental results for specific examples of NBO demonstrate the benefits of incorporating curvature information into the optimization process for bilevel problems. However, many additional benefits and extensions could be explored, such as the exploitation of parallel and distributed computation, and the integration of noise reduction techniques.

## Acknowledgements

Authors listed in alphabetical order. This work is supported by the National Natural Science Foundation of China (12431011, 12371301, 12371305, 12222106, 12326605), Natural Science Foundation for Distinguished Young Scholars of Gansu Province (22JR5RA223), Guangdong Basic and Applied Basic Research Foundation (No. 2022B1515020082) and the Longhua District Science and Innovation Commission Project Grants of Shenzhen (Grant No.20250113G43468522). We thank the anonymous reviewers for their valuable comments and constructive suggestions on this work.

## Impact Statement

This paper presents work whose goal is to advance the field of Machine Learning. There are many potential societal consequences of our work, none which we feel must be specifically highlighted here.

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

## A. Appendix

The appendix is organized as follows:

- More related work is provided in Section B.

- Experimental details and additional experiments are provided in Section C.

- The initialization strategy of the proposed algorithms is provided in Section D.

- Detailed proofs of the main theorems are provided in Section E.

## B. More Related Works

**Bilevel optimization.** Bilevel optimization addresses the challenges associated with nested optimization structures commonly encountered in various machine learning applications, as discussed in recent survey papers (Liu et al., 2021; Zhang et al., 2024). Although numerous methods have emerged, the numerical computation of bilevel optimization remains a significant challenge, even when the lower-level problem has a unique solution (Pedregosa, 2016; Ghadimi & Wang, 2018; Kwon et al., 2023). Assuming further that the Hessian of the lower-level objective function *w.r.t* the lower-level variables is invertible, the hypergradient is well-defined and can be derived from the implicit function theorem. Recently, several strategies have been proposed to solve bilevel optimization. For instance, the iterative differentiation (ITD)-based approach (Maclaurin et al., 2015; Franceschi et al., 2018; Grazzi et al., 2020; Ji et al., 2021) estimates the Jacobian of the lower-level solution map by differentiating the steps used to compute an approximation of the lower-level solution. The approximate implicit differentiation (AID)-based approach (Ghadimi & Wang, 2018; Chen et al., 2021; Ji et al., 2021; 2022; Arbel & Mairal, 2022a; Dagréou et al., 2022; Hong et al., 2023) is directly based on equation (2). It performs several (stochastic) gradient descent steps in the lower-level problem, followed by the estimation of the Hessian inverse-vector product, which can be computed using Neumann approximations (Chen et al., 2021; Hong et al., 2023), solving a linear system (Pedregosa, 2016; Ji et al., 2021), or solving a quadratic programming problem (Arbel & Mairal, 2022a; Dagréou et al., 2022).

The ITD- and AID-based methods involve numerous Hessian- and Jacobian-vector products, which can be efficiently computed and stored using existing automatic differentiation packages (Pearlmutter, 1994; Dagréou et al., 2024). Another class of methods, which rely solely on the first-order gradients of the upper- and lower-level objective functions, is based on the value function reformulation (Ye & Zhu, 1995) and is referred to as the value function-based approach (Liu et al., 2022; Kwon et al., 2023; Chen et al., 2023a; Liu et al., 2023a; Kwon et al., 2024). From the hypergradient perspective, all of the aforementioned gradient-based bilevel optimization algorithms are inexact hypergradient methods. The distinction between these methods lies in whether the hypergradient approximation is directly derived from equation (2). For instance, the AID-based approach is directly based on equation (2), whereas the ITD-based and value function-based approaches are not. However, what these methods do is to approximate the hypergradient. For example, Proposition 2 in Ji et al. (2021) provides the explicit form of the ITD-based hypergradient estimate, while Lemma 3.1 in Kwon et al. (2023) offers a hypergradient estimate for the value function-based approach. In stochastic bilevel optimization, various stochastic techniques (e.g., momentum and variance reduction) from single-level optimization have been employed to improve the convergence rate (Yang et al., 2021; Dagréou et al., 2022; Khanduri et al., 2021; Chen et al., 2024c; Dagréou et al., 2024; Chu et al., 2024; Huang, 2024).

Although our paper focuses on the lower-level strongly convex case, the lower-level non-strongly convex case is also of great interest. In some works, the hypergradient can still be approximated in non-strongly convex case, by combining the pseudo inverse of the Hessian (Arbel & Mairal, 2022b; Xiao et al., 2023) or aggregation functions (Liu et al., 2023b). Then this approximate hypergradient can be used to iteratively update the upper-level variable. There are also works that employ penalty function methods, including but not limited to Kwon et al. (2024); Shen & Chen (2023); Chen et al. (2024a).

## C. Experimental Details and Additional Experiments

Our experiments were conducted using the Bilevel Optimization Benchmark framework (Dagréou et al., 2022) and the Benchopt library (Moreau et al., 2022). All experiments were performed on a system equipped with an Intel(R) Xeon(R)

Gold 5218R CPU running at 2.10 GHz and an NVIDIA H100 GPU with 80 GB of memory. Each experiment was repeated ten times.

## C.1. Synthetic Data

For the synthetic dataset, we use 16,000 training samples and 4,000 validation samples. The dimension size is set to $p = 50$. For both AmIGO and NBO, the outer step size is set to 1, and the inner step size is set to 0.03. Both algorithms are deterministic, employing full-batch updates. The experimental results for $r' = 0.5$ and $r' = 2$ are presented in Figures 6 and 7, respectively.

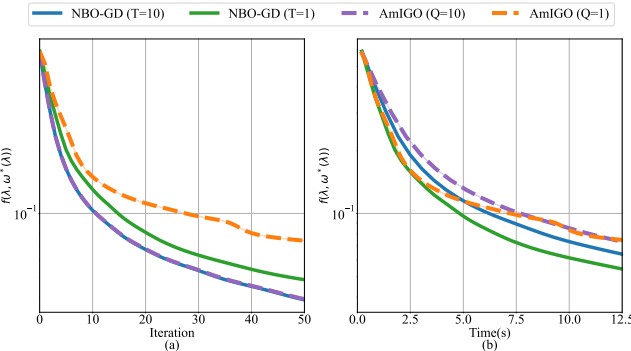

*Figure 6.* Experimental results on synthetic data with $r' = 0.5$.

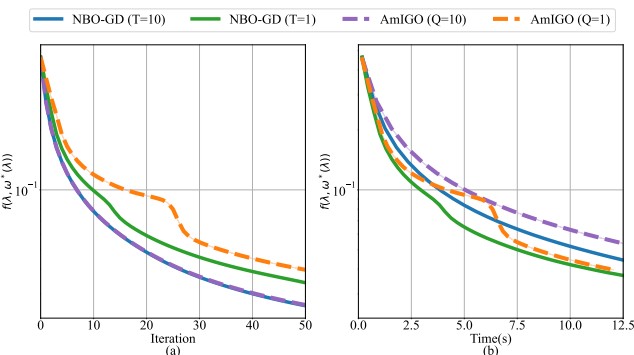

*Figure 7.* Experimental results on synthetic data with $r' = 2$.

The results align with our discussion in Section 4.1. Figures 6 and 7 demonstrate that NBO-GD maintains strong performance even when the Newton direction is approximated with $T = 1$. In contrast, AmIGO's performance deteriorates significantly when only a single step of gradient descent is used. When considering runtime, the results show that NBO-GD ($T = 1$) outperforms other methods. Moreover, we evaluate the scalability of NBO on synthetic data while progressively increasing the problem dimension in Figure 8. We record the time it takes to achieve $(f(\lambda^k, \omega^*(\lambda^k)) - \Phi^*)/(f(\lambda^0, \omega^*(\lambda^0)) - \Phi^*) < 10^{-3}$ when the dimension increasing. The results after 5000 iterations is used as $\Phi^*$.

## C.2. Hyperparameter Optimization and Data Hyper-Cleaning

**Datasets.** We provide detailed information regarding the datasets used in our experiments. For IJCNN1[1], we employ 49,990 training samples and 91,701 validation samples. For Covtype[2], we utilize 371,847 training samples, 92,962 validation samples, and 116,203 testing samples. For MNIST[3] and FashionMNIST[4], we use 20,000 training samples, 5,000 validation

---

[1]https://www.csie.ntu.edu.tw/ cjlin/libsvmtools/datasets/binary.html

[2]https://scikit-learn.org/stable/modules/generated/sklearn.datasets.fetch_covtype.html

[3]http://yann.lecun.com/exdb/mnist/

[4]https://github.com/zalandoresearch/fashion-mnist

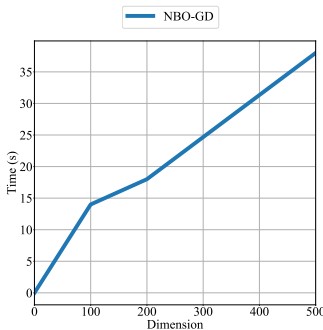

*Figure 8.* Scalability of NBO to large dimensional problems on synthetic data.

samples, and 10,000 testing samples, and we corrupt the labels of these two datasets with a probability of $p' = 0.5$. In terms of computational frameworks, we use JAX (Bradbury et al., 2018) for MNIST, FashionMNIST, and Covtype. For IJCNN1, we use Numba (Lam et al., 2015), as it demonstrates faster performance compared to JAX for this dataset.

**Algorithm Settings.** We provide a detailed description of the batch size, step size, and other settings for the algorithms compared in Section 4.2. The batch size for all algorithms is set to 64, except for NSBO-SGD and SHINE. For NSBO-SGD, since the size of $B_2^k$ is significantly larger than that of other batches in theory, we set $|B_2^k| = 256$, while the other batches remain at 64. For SHINE, we set the batch size to 256, as it is a deterministic algorithm by design. In the benchmark, the double-loop algorithms, including AmIGO, StocBiO, and F2SA, are configured with 10 inner loop iterations, and we set the number of iterations for BFGS in SHINE to 5. The step sizes are tuned via grid search, following a method similar to (Dagréou et al., 2022). The grid search procedure is described in detail below. For algorithms that employ a decreasing step size, the step sizes we search for correspond to their initial values. For SHINE, the inner step size we search for is related to the initial step size of the strong-Wolfe line search. The outer step size is computed as $\frac{\text{inner step size}}{\text{outer ratio}}$.

- **IJCNN1**: The inner step size is chosen from 6 values between $2^{-5}$ and 1, spaced on a logarithmic scale. The outer ratio is chosen in $\left\{ 10^{-2}, 10^{-1.5}, 10^{-1}, 10^{-0.5}, 1 \right\}$.

- **Covtype**: The inner step size is chosen from 7 values between $2^{-6}$ and 1, spaced on a logarithmic scale. The outer ratio is chosen from 4 values between $10^{-2}$ and 10, spaced on a logarithmic scale.

- **MNIST and FashionMNIST**: The inner step size is chosen from 4 values between $10^{-3}$ and 1, spaced on a logarithmic scale. The outer ratio is chosen from 4 values between $10^{-6}$ and $10^{-3}$, spaced on a logarithmic scale.

Moreover, for MA-SABA and MA-NSBO-SAGA, we set the moving average coefficient to 0.99 for MNIST and FashionMNIST, 0.9 for IJCNN1, 0.8 for Covtype. Last but not least, we do not follow the initialization in Algorithm 2 and Algorithm 4. Instead, we set the initial points for our framework in the same way as other algorithms in the benchmark. Other settings not mentioned are kept consistent with the benchmark.

**Additional results.** We present the additional experimental results for Covtype and FashionMNIST. In Figure 9 (a) , the additional results of hyperparameter optimization on Covtype are shown. In Figure 9 (b), the additional results of data hyper-cleaning on FashionMNIST are displayed. These results demonstrate the effectiveness of our framework under different implementations. In Figure 9 (b), the performance of MA-SABA and MA-NSBO-SAGA is similar, likely because MA-SABA is fast enough on FashionMNIST, leaving little room for further acceleration.

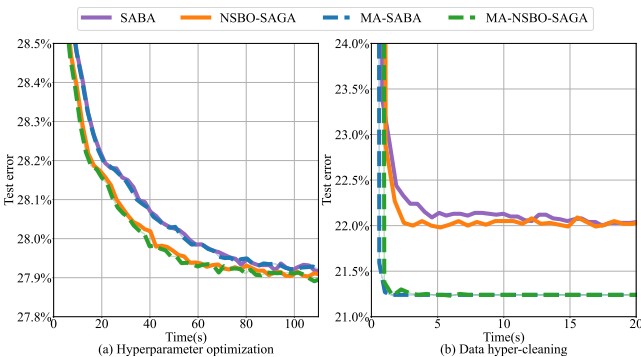

*Figure 9.* Comparison between NBO framework and other algorithms equipped with variance reduction and moving average technique. **Left:** Hyperparameter optimization on Covtype; **Right:** Data hyper-cleaning on FashionMNIST.

## D. Initialization Strategy of the Proposed Algorithms

In this section, we introduce two initialization boxes for our proposed algorithms. These initializations ensure that the initial points $y^0$, $u^0$ are close to $y^*(x^0)$, $u^*(x^0)$, respectively, thereby guaranteeing that $y^k$, $u^k$ remain within a neighborhood of $y^*(x^k)$, $u^*(x^k)$ for all $k = 0, \cdots, K$ with a controlled step size $\alpha_k$. It is important to note that the primary purpose of these initialization requirements is to ensure theoretical validity. In practice, strict adherence to these constraints is not necessary, as the proposed initialization process does not introduce significant computational overhead. Some constants in this section, such as $r_u$ and $L_1$, are defined in (29).

---

- Choose $x^0, y^{0,0}, u^{0,0}$, stepsize $\beta_0 \leq \frac{1}{L_{f,1}}$.

- Take $N_0 \geq 2\log \frac{\min\left\{\frac{\mu}{2\sqrt{L_1}}, \frac{\mu^2}{2L_{g,2}}\right\}}{\|\nabla g(x^0, y^{0,0})\|} \Big/ \log(1 - \beta_0\mu) = \Theta(\kappa \log \kappa)$ and compute $y^0 = y^{N_0,0}$:

$$\mathbf{for} \ n = 0, 1, \ldots, N_0 - 1$$
$$y^{n+1,0} = y^{n,0} - \beta_0 \nabla_2 g(x^0, y^{n,0})$$
$$\mathbf{end \ for}$$

- Take $Q_0 \geq \log \frac{\min\left\{\frac{5L_1}{4L_{g,2}}, \frac{\sqrt{L_1}}{2\mu}\right\}}{\|u^{0,0}\| + \frac{L_1}{\mu^2}\|\nabla g(x^0, y^0)\| + r_u} \Big/ \log(1 - \beta_0\mu) = \Theta(\kappa \log \kappa)$ and compute $u^0 = u^{Q_0,0}$:

$$\mathbf{for} \ q = 0, 1, \ldots, Q_0 - 1$$
$$u^{q+1,0} = u^{q,0} - \beta_0\big(\nabla_{22}^2 g(x^0, y^0)u^{q,0} - \nabla_2 f(x^0, y^0)\big)$$
$$\mathbf{end \ for}$$

- Output $x^0, y^0, u^0$.

BOX 1: Initialization of Algorithm 2.

---

- Choose $x^0, y^{0,0}, u^{0,0}$, stepsize $\beta_0 \leq \frac{1}{L_{f,1}}$, batch size $|B_{0,n}|, |B_{0,q}| \geq \Theta(\kappa^5), |B'_{0,q}| \geq \Theta(1)$ according to (24).

- Take $N_0 \geq \log \frac{\min\left\{\frac{\mu^2}{8L_1}, \frac{\mu^4}{40rL_{g,2}^2}\right\}}{\|\nabla g(x^0,y^{0,0})\|^2} / \log(1 - \beta_0\mu) = \Theta(\kappa \log \kappa)$ and compute $y^0 = y^{N_0,0}$:

$$\textbf{for } n = 0, 1, \ldots, N_0 - 1$$
$$y^{n+1,0} = y^{n,0} - \beta_0 \nabla_2 G(x^0, y^{n,0}; B_{0,n})$$
$$\textbf{end for}$$

- Take $Q_0 \geq \log \frac{\min\left\{\frac{2L_1^2}{5rL_{g,2}^2}, \frac{L_1}{2\mu^2}\right\}}{4\left(\|u^{0,0}\| + \frac{L_1}{\mu^2}\|\nabla g(x^0,y^0)\| + r_u\right)^2} / \log(1 - \beta_0\mu/2) = \Theta(\kappa \log \kappa)$ and compute $u^0 = u^{Q_0,0}$:

$$\textbf{for } q = 0, 1, \ldots, Q_0 - 1$$
$$u^{q+1,0} = u^{q,0} - \beta_0\left(\nabla_{22}^2 G(x^0, y^0; B_{0,q})u^{q,0} - \nabla_2 f(x^0, y^0; B'_{0,q})\right)$$
$$\textbf{end for}$$

- Output $x^0, y^0, u^0$.

BOX 2: Initialization of Algorithm 4.

The following two lemmas describe the properties of the output sequences from BOX 1 and BOX 2, respectively.

**Lemma D.1.** *Suppose Assumption 3.1 holds, then $x^0, y^0, u^0$ in BOX 1 satisfy $\|y^0 - y^*(x^0)\| \leq \min\left\{\frac{\mu}{2L_{g,2}}, \frac{1}{2\sqrt{L_1}}\right\}$, $\|u^0 - u^*(x^0)\| \leq \min\left\{\frac{5L_1}{2L_{g,2}}, \frac{\sqrt{L_1}}{\mu}\right\}$.*

*Proof.* For $y^{n+1,0} = y^{n,0} - \beta_0 \nabla_2 g(x^0, y^{n,0})$, we have

$$\left\|y^{n+1,0} - y^*(x^0)\right\|^2 = \left\|y^{n,0} - \beta_0 \nabla_2 g(x^0, y^{n,0}) - y^*(x^0)\right\|^2$$
$$= \|y^{n,0} - y^*(x^0)\|^2 - 2\beta_0 \left\langle y^{n,0} - y^*(x^0), \nabla_2 g(x^0, y^{n,0})\right\rangle + \beta_0^2 \|\nabla_2 g(x^0, y^{n,0})\|^2.$$

Since $\nabla_2 g(x^0, y^*(x^0)) = 0$, Assumption 3.1 and Theorem 2.1.12 in (Nesterov, 2018) implies that

$$\left\langle y^{n,0} - y^*(x^0), \nabla_2 g(x^0, y^{n,0})\right\rangle = \left\langle y^{n,0} - y^*(x^0), \nabla_2 g(x^0, y^{n,0}) - \nabla_2 g(x^0, y^*(x^0))\right\rangle$$
$$\geq \frac{\mu L_{g,1}}{\mu + L_{g,1}}\|y^{n,0} - y^*(x^0)\|^2 + \frac{1}{\mu + L_{g,1}}\|\nabla_2 g(x^0, y^{n,0})\|^2.$$

Since $\beta_0 \leq \frac{1}{L_{g,1}} \leq \frac{2}{\mu + L_{g,1}}$, we have

$$\left\|y^{n+1,0} - y^*(x^0)\right\|^2 \leq \left(1 - 2\beta_0 \frac{\mu L_{g,1}}{\mu + L_{g,1}}\right)\|y^{n,0} - y^*(x^0)\|^2 \leq (1 - \beta_0\mu)\|y^{n,0} - y^*(x^0)\|^2. \tag{19}$$

Then we can deduce that

$$\left\|y^{N_0,0} - y^*(x^0)\right\| \leq (1 - \beta_0\mu)^{N_0/2}\|y^{0,0} - y^*(x^0)\| \leq (1 - \beta_0\mu)^{N_0/2}\frac{1}{\mu}\|\nabla g(x^0, y^{0,0})\| \leq \min\left\{\frac{1}{2\sqrt{L_1}}, \frac{\mu}{2L_{g,2}}\right\},$$

where the second inequality follows from

$$\|y^{0,0} - y^*(x^0)\| \leq \frac{1}{\mu}\|\nabla g(x^0, y^{0,0}) - \nabla g(x^0, y^*(x^0))\| = \frac{1}{\mu}\|\nabla g(x^0, y^{0,0})\| \tag{20}$$

by $\mu$-strong convexity of $g(x, \cdot)$ and the optimality of $y^*(x^0)$. Next, recall that $u^*(x, y) = [\nabla^2_{22} g(x, y)]^{-1} \nabla_2 f(x, y)$, the difference between $u^*(x)$ and $u^*(x, y)$ is

$$
\begin{aligned}
\mu \| u^*(x, y) - u^*(x) \| &\leq \| \nabla^2_{22} g(x, y)(u^*(x, y) - u^*(x)) \| \\
&\leq \| \nabla_2 f(x, y) - \nabla_2 f(x, y^*(x)) \| + \| u^*(x) \| \| \nabla^2_{22} g(x, y^*(x)) - \nabla^2_{22} g(x, y) \| \\
&\leq (L_{f,1} + r_u L_{g,2}) \| y - y^*(x) \| = L_1 \| y - y^*(x) \|,
\end{aligned}
\tag{21}
$$

where $\| u^*(x) \| \leq \frac{L_{f,0}}{\mu} = r_u$ by definition, the second inequality is because $\nabla^2_{22} g(x, y^*(x)) u^*(x) = \nabla_2 f(x, y^*(x))$ and $\nabla^2_{22} g(x, y) u^*(x, y) = \nabla_2 f(x, y)$. For $u^{q+1,0} = u^{q,0} - \beta_0 \big( \nabla^2_{22} g(x^0, y^0) u^{q,0} - \nabla_2 f(x^0, y^0) \big)$, by $\mu$-strong convexity of $g(x, \cdot)$ and $\beta_0 \leq \frac{1}{L_{g,1}}$, we have

$$
\begin{aligned}
\| u^{q+1,0} - u^*(x^0, y^0) \| &= \| u^{q,0} - \beta_0 \big( \nabla^2_{22} g(x^0, y^0) u^{q,0} - \nabla_2 f(x^0, y^0) \big) - u^*(x^0, y^0) \| \\
&\leq \big\| I - \beta_0 \nabla^2_{22} g(x^0, y^0) \big\|_{\text{op}} \big\| u^{q,0} - u^*(x^0, y^0) \big\| \leq (1 - \beta_0 \mu) \| u^{q,0} - u^*(x^0, y^0) \|.
\end{aligned}
\tag{22}
$$

So we can deduce that

$$
\begin{aligned}
\| u^{Q_0,0} - u^*(x^0) \| &= \| u^{Q_0,0} - u^*(x^0, y^0) \| + \| u^*(x^0, y^0) - u^*(x^0) \| \\
&\overset{(22),(21)}{\leq} (1 - \beta_0 \mu)^{Q_0} \| u^{0,0} - u^*(x^0, y^0) \| + \frac{L_1}{\mu} \| y^0 - y^*(x^0) \| \\
&\leq (1 - \beta_0 \mu)^{Q_0} \big( \| u^{0,0} \| + \| u^*(x^0, y^0) \| \big) + \frac{L_1}{\mu} \min \big\{ \frac{1}{2\sqrt{L_1}}, \frac{\mu}{2 L_{g,2}} \big\} \leq \min \big\{ \frac{5 L_1}{2 L_{g,2}}, \frac{\sqrt{L_1}}{\mu} \big\},
\end{aligned}
$$

where

$$
\| u^*(x^0, y^0) \| \leq \| u^*(x^0, y^0) - u^*(x^0) \| + \| u^*(x^0) \| \overset{(21)}{\leq} \frac{L_1}{\mu} \| y^0 - y^*(x^0) \| + r_u \leq \frac{L_1}{\mu^2} \nabla g(x^0, y^0) + r_u.
\tag{23}
$$

Hence we complete the proof. $\qquad \square$

**Lemma D.2.** *Suppose Assumptions 3.1, 3.5 hold, when we take the bachsize*

$$
\begin{aligned}
|B_0| &\geq \max \Big\{ \frac{8 \beta_0 \sigma^2_{g,2}}{\mu}, \frac{\sigma^2_{g,1} \beta_0}{\mu \min \{ \mu^2/(40 r L^2_{g,2}), 1/(8 L_1) \}}, \frac{32 \beta_0 \sigma^2_{g,1} \big( \frac{L_1}{\mu^2} \| \nabla g(x^0, y^0) \| + r_u \big)^2}{\mu \min \{ L^2_1/(5 r L^2_{g,2}), L_1/(4 \mu^2) \}} \Big\} = \Theta(\kappa^5), \\
|B_0'| &\geq \Theta(1) \geq \frac{16 \beta_0 \sigma^2_{f,1}}{\mu \min \{ L^2_1/(5 r L^2_{g,2}), L_1/(4 \mu^2) \}},
\end{aligned}
\tag{24}
$$

*then $x^0, y^0, u^0$ in BOX 2 satisfy $\mathbb{E} \big[ \| y^0 - y^*(x^0) \|^2 \big] \leq \min \big\{ \frac{\mu^2}{20 r L^2_{g,2}}, \frac{1}{4 L_1} \big\}$ and $\mathbb{E} \big[ \| u^0 - u^*(x^0) \|^2 \big] \leq \min \big\{ \frac{4 L^2_1}{5 r L^2_{g,2}}, \frac{L_1}{\mu^2} \big\}$.*

*Proof.* Similar to Lemma D.1, we can deduce that

$$
\| y^{n,0} - \beta_0 \nabla_2 g(x^0, y^{n,0}) - y^*(x^0) \|^2 \leq (1 - \mu \beta_0) \| y^{n,0} - y^*(x^0) \|^2.
$$

Then denote a conditional expectation $\mathbb{E}_y^n := \mathbb{E} \big[ \cdot | x^0, y^{n,0} \big]$, for $y^{n+1,0} = y^{n,0} - \beta_0 \nabla_2 G(x^0, y^{n,0}; B_0)$, we have

$$
\begin{aligned}
\mathbb{E}_y^n \big[ \| y^{n+1,0} - y^*(x^0) \|^2 \big] &= \| y^{n,0} - \beta_0 \nabla_2 g(x^0, y^{n,0}) - y^*(x^0) \|^2 + \beta_0^2 \mathbb{E}_y^n \big[ \| \nabla_2 g(x^0, y^{n,0}) - \nabla_2 G(x^0, y^{n,0}; B_0) \|^2 \big] \\
&\leq (1 - \mu \beta_0) \| y^{n,0} - y^*(x^0) \|^2 + \beta_0^2 \frac{\sigma^2_{g,1}}{|B_0|},
\end{aligned}
\tag{25}
$$

take expectation and by induction we obtain

$$
\mathbb{E} \big[ \| y^{N_0,0} - y^*(x^0) \|^2 \big] \leq (1 - \mu \beta_0)^{N_0} \| y^{0,0} - y^*(x^0) \|^2 + \beta_0 \frac{\sigma^2_{g,1}}{\mu |B_0|} \leq \min \big\{ \frac{\mu^2}{20 r L^2_{g,2}}, \frac{1}{4 L_1} \big\},
$$

where $\|y^{0,0} - y^*(x^0)\| \leq \frac{1}{\mu}\|\nabla g(x^0, y^{0,0})\|$ from (20). Next, from (22) we have

$$\left\|u^{q,0} - \beta_0\big(\nabla_{22}^2 g(x^0, y^0)u^{q,0} - \nabla_2 f(x^0, y^0)\big) - u^*(x^0, y^0)\right\| \leq (1 - \beta_0\mu)\|u^{q,0} - u^*(x^0, y^0)\|,$$

denote $\mathbb{E}_u^q := \mathbb{E}\left[\cdot\,|x^0, y^0, u^{q,0}\right]$, we have

$$\mathbb{E}_u^q\left[\|u^{q+1,0} - u^*(x^0, y^0)\|^2\right] = \left\|u^{q,0} - \beta_0\big(\nabla_{22}^2 g(x^0, y^0)u^{q,0} - \nabla_2 f(x^0, y^0)\big) - u^*(x^0, y^0)\right\|^2$$
$$+ \beta_0^2 \mathbb{E}_u^q\left[\|\nabla_{22}^2 g(x^0, y^0)u^{q,0} - \nabla_2 f(x^0, y^0) - \nabla_{22}^2 G(x^0, y^0; B_0)u^{q,0} + \nabla_2 F(x^0, y^0; B_0)\|^2\right]$$

$$\leq (1 - \beta_0\mu)^2\|u^{q,0} - u^*(x^0, y^0)\|^2 + 2\beta_0^2\frac{\sigma_{f,1}^2}{|B_0'|} + 4\beta_0^2\frac{\sigma_{g,2}^2}{|B_0|}\|u^{q,0} - u^*(x^0, y^0)\|^2 + 4\beta_0^2\frac{\sigma_{g,2}^2}{|B_0|}\|u^*(x^0, y^0)\|^2,$$

$$\leq (1 - \beta_0\mu/2)\|u^{q,0} - u^*(x^0, y^0)\|^2 + 2\beta_0^2\frac{\sigma_{f,1}^2}{|B_0'|} + 4\beta_0^2\frac{\sigma_{g,1}^2}{|B_0|}\|u^*(x^0, y^0)\|^2,$$

where the last inequality is because $|B_0| \geq \frac{8\beta_0\sigma_{g,2}^2}{\mu}$. Then taking conditional expetation $\mathbb{E}' = \mathbb{E}[\cdot|x^0, y^0]$ on both side of the inequality and by induction,

$$\mathbb{E}'\left[\|u^{Q_0,0} - u^*(x^0, y^0)\|^2\right] \leq (1 - \beta_0\mu/2)^{Q_0}\|u^{0,0} - u^*(x^0, y^0)\|^2 + 4\beta_0\frac{\sigma_{f,1}^2}{\mu|B_0'|} + 8\beta_0\frac{\sigma_{g,1}^2}{\mu|B_0|}\|u^*(x^0, y^0)\|^2.$$

Combining with (21) and (23), we obtain

$$\mathbb{E}'\left[\|u^{Q_0,0} - u^*(x^0)\|^2\right] \leq 2\mathbb{E}'\left[\|u^{Q_0,0} - u^*(x^0, y^0)\|^2\right] + 2\|u^*(x^0) - u^*(x^0, y^0)\|^2$$

$$\leq 2(1 - \beta_0\mu/2)^{Q_0}\big(\|u^{0,0}\| + \|u^*(x^0, y^0)\|\big)^2 + 8\beta_0\frac{\sigma_{f,1}^2}{\mu|B_0'|} + 16\beta_0\frac{\sigma_{g,1}^2}{\mu|B_0|}\|u^*(x^0, y^0)\|^2 + \frac{2L_1^2}{\mu^2}\mathbb{E}\left[\|y^0 - y^*(x^0)\|^2\right]$$

$$\leq 2(1 - \beta_0\mu/2)^{Q_0}\big(\|u^{0,0}\| + \frac{L_1}{\mu^2}\|\nabla g(x^0, y^0)\| + r_u\big)^2$$

$$+ 8\beta_0\frac{\sigma_{f,1}^2}{\mu|B_0'|} + 16\beta_0\frac{\sigma_{g,1}^2}{\mu|B_0|}\big(\frac{L_1}{\mu^2}\|\nabla g(x^0, y^0)\| + r_u\big)^2 + \frac{2L_1^2}{\mu^2}\mathbb{E}\left[\|y^0 - y^*(x^0)\|^2\right] \leq \min\big\{\frac{4L_1^2}{5rL_{g,2}^2}, \frac{L_1}{\mu^2}\big\},$$

then we complete the proof. $\qquad\square$

*Remark* D.3. From Lemma D.1, we deduce that the initialization in BOX 1 requires $O(\kappa\log\kappa)$ gradient computations and Hessian-vector products. Similarly, Lemma D.2 indicates that the initialization in BOX 2 has a gradient complexity of $O(\kappa^6\log\kappa)$ for $g$, $O(\kappa\log\kappa)$ for $f$, and a Hessian-vector product complexity of $O(\kappa^6\log\kappa)$.

## E. Detailed Proofs

In this section of the Appendix, we provide detailed proofs of Theorem 3.2, Theorem 3.7, and Remark 3.4. Recall that

$$d_x^k := \nabla_1 f(x^k, y^k) - \nabla_{12}^2 g(x^k, y^k)u^k, \ D_x^k := \nabla_1 F(x^k, y^k; B_3^k) - \nabla_{12}^2 G(x^k, y^k; B_4^k)u^k, \tag{26}$$

$$d_y^k := \nabla_2 g(x^k, y^k), \ D_y^k = \nabla_2 G(x^k, y^k; B_2^k), \tag{27}$$

$$d_u^k := \nabla_{22}^2 g(x^k, y^k)u^k - \nabla_2 f(x^k, y^k), \ D_u^k := \nabla_{22}^2 G(x^k, y^k; B_1^k)u^k - \nabla_2 F(x^k, y^k; B_3^k). \tag{28}$$

For convenience, we further define

$$r_u := L_{f,0}/\mu, \ L_u := (L_{f,1} + L_{g,2}r_u)(1 + L_{g,1}/\mu), \ L_1 := L_{f,1} + L_{g,2}r_u, \ L_2 := C_{f,0} + L_{g,1}r_u, \tag{29}$$

$$w^*(x^k, y^k, u^k) = u^k - [\nabla_{22}^2 g(x^k, y^k)]^{-1}\nabla_2 f(x^k, y^k),$$

$$d_v^{t,k} := \nabla_{22}^2 g(x^k, y^k)v^{t,k} - \nabla_2 g(x^k, y^k), \ D_v^{t,k} := \nabla_{22}^2 G(x^k, y^k; B_1^{t,k})v^{t,k} - \nabla_2 G(x^k, y^k; B_2^k), \tag{30}$$

$$d_w^{t,k} := \nabla_{22}^2 g(x^k, y^k)w^{t,k} - \nabla_{22}^2 g(x^k, y^k)u^k + \nabla_2 f(x^k, y^k),$$

$$D_w^{t,k} := \nabla_{22}^2 G(x^k, y^k; B_1^{t,k})w^{t,k} - \nabla_{22}^2 G(x^k, y^k; B_1^k)u^k + \nabla_2 F(x^k, y^k; B_3), \tag{31}$$

$$\mathbb{E}^k[\cdot] := \mathbb{E}[\cdot|x^k, y^k, u^k], \ \mathbb{E}^{t,k}[\cdot] = \mathbb{E}[\cdot|x^k, y^k, u^k, v^{t,k}, w^{t,k}],$$

$$S_{g,1} := \frac{\sigma_{g,1}^2}{|B_2^k|}, \ S_1^{g,2} := \frac{\sigma_{g,2}^2}{|B_1^{t,k}|}, \ S_2^{g,2} := \frac{\sigma_{g,2}^2}{|B_1^k|}, \ S_3^{g,2} := \frac{\sigma_{g,2}^2}{|B_4^k|}, \ S_{f,1} := \frac{\sigma_{f,1}^2}{|B_3^k|}. \tag{32}$$

The following lemma is fundamental and will be repeatedly invoked throughout the proof.

**Lemma E.1.** *Suppose Assumption 3.1 holds, we have*

$$\|y^*(x') - y^*(x)\| \le \frac{L_{g,1}}{\mu}\|x' - x\| \quad and \quad \|u^*(x') - u^*(x)\| \le \frac{L_u}{\mu}\|x' - x\|. \tag{33}$$

*Proof.* From the definition of $u^*(x)$, by Assumption 3.1, we have $\|u^*(x)\| \le \|[\nabla_{22}^2 g(x, y^*(x))]^{-1}\|_{\text{op}}\|\nabla_2 f(x, y^*(x))\| \le \frac{L_{f,0}}{\mu} = r_u$. Due to the optimality of $y^*(x)$, we have $\nabla_2 g(x, y^*(x)) = 0$. Since $g(x, \cdot)$ is $\mu$-strongly convex and $g(\cdot, y)$ is $L_{g,1}$-smooth, we can deduce that

$$\mu\|y^*(x) - y^*(x')\| \le \|\nabla_2 g(x, y^*(x)) - \nabla_2 g(x, y^*(x'))\|$$
$$= \|\nabla_2 g(x, y^*(x')) - \nabla_2 g(x', y^*(x'))\| \le L_{g,1}\|x - x'\|,$$

which implies the first inequality in (33). Next, by the strong convexity of $g(x, \cdot)$ and smoothness of $g, f$, we have

$$\|\nabla_{22}^2 g(x, y^*(x))(u^*(x) - u^*(x'))\| \ge \mu\|u^*(x) - u^*(x')\|, \tag{34}$$

and

$$\left\|\nabla_2 f(x, y^*(x)) - \nabla_2 f(x', y^*(x')) + \left[\nabla_{22}^2 g(x', y^*(x')) - \nabla_{22}^2 g(x, y^*(x))\right]u^*(x')\right\|$$
$$\le \left(L_{f,1} + L_{g,2}r_u\right)\left(\|x' - x\| + \|y^*(x') - y^*(x)\|\right). \tag{35}$$

By the definition of $u^*(x)$, the following equality holds

$$\nabla_{22}^2 g(x, y^*(x))(u^*(x) - u^*(x')) = \nabla_2 f(x, y^*(x)) - \nabla_{22}^2 g(x, y^*(x))u^*(x')$$
$$+ \nabla_{22}^2 g(x', y^*(x'))u^*(x') - \nabla_2 f(x', y^*(x')).$$

Taking the norm on both sides of the above inequality and combining (34), (35), we can deduce the second inequlity in (33). $\square$

### E.1. Detailed Proof of Theorem 3.2 and Remark 3.4

We first establish the descent of $\Phi(x^k)$, $\|y^k - y^*(x^k)\|^2$ and $\|u^k - u^*(x^k)\|^2$ respectively.

**Lemma E.2.** *Suppose Assumption 3.1 holds, the sequences generated by Algorithm 2 satisfy*

$$\Phi(x^{k+1})$$
$$\le \Phi(x^k) - \frac{\alpha_k}{2}\|\nabla\Phi(x^k)\|^2 - \left(\frac{\alpha_k}{2} - \frac{L_\Phi \alpha_k^2}{2}\right)\|d_x^k\|^2 + L_1^2\alpha_k\|y^k - y^*(x^k)\|^2 + L_{g,1}^2\alpha_k\|u^k - u^*(x^k)\|^2. \tag{36}$$

*Proof.* From Lemma 2.2 in (Ghadimi & Wang, 2018) we know that $\Phi(x)$ is $L_\Phi$-smooth, and $L_\Phi$ is defined in (17), then we have

$$\Phi(x^{k+1}) \le \Phi(x^k) + \langle\nabla\Phi(x^k), x^{k+1} - x^k\rangle + \frac{L_\Phi}{2}\|x^{k+1} - x^k\|^2$$

$$= \Phi(x^k) - \alpha_k\langle\nabla\Phi(x^k), d_x^k\rangle + \frac{L_\Phi\alpha_k^2}{2}\|d_x^k\|^2$$

$$= \Phi(x^k) + \frac{\alpha_k}{2}\|\nabla\Phi(x^k) - d_x^k\|^2 - \frac{\alpha_k}{2}\|\nabla\Phi(x^k)\|^2 - \frac{\alpha_k}{2}\|d_x^k\|^2 + \frac{L_\Phi\alpha_k^2}{2}\|d_x^k\|^2, \tag{37}$$

where

$$\|\nabla\Phi(x^k) - d_x^k\| \le \|\nabla_1 f(x^k, y^*(x^k)) - \nabla_1 f(x^k, y^k)\| + \|[\nabla_{12}^2 g(x^k, y^k) - \nabla_{12}^2 g(x^k, y^*(x^k))]u^*(x^k)\|$$
$$+ \|\nabla_{12}^2 g(x^k, y^k)[u^k - u^*(x^k)]\|$$
$$\le (L_{f,1} + L_{g,2}r_u)\|y^k - y^*(x^k)\| + L_{g,1}\|u^k - u^*(x^k)\|. \tag{38}$$

Then combining (38) with (37), we can obtain the desired result. $\square$

**Lemma E.3.** *Suppose Assumption 3.1 holds. When we take $T \geq \Theta(\kappa)$, the sequences generated by Algorithm 2 satisfy*

$$\|y^{k+1} - y^*(x^{k+1})\| \leq \frac{L_{g,2}}{2\mu}\|y^*(x^k) - y^k\|^2 + \frac{L_{g,1}}{\mu}\|x^{k+1} - x^k\| + \frac{1}{4}\|y^k - y^*(x^k)\|, \tag{39}$$

$$\|u^{k+1} - u^*(x^{k+1})\| \leq \frac{1}{4}\|u^k - u^*(x^k)\| + \frac{5L_1}{4\mu}\|y^k - y^*(x^k)\| + \frac{L_u}{\mu}\|x^{k+1} - x^k\|. \tag{40}$$

*Proof.* From the iteration of Algorithm 2 we know that

$$y^k - v^{t+1,k} = y^k - \left[I - \gamma_k \nabla_{22}^2 g(x^k, y^k)\right] v^{t,k} - \gamma_k \nabla_2 g(x^k, y^k).$$

Combining $\nabla_2 g(x^k, y^*(x^k)) = 0$, we have

$$y^k - v^{t+1,k} - y^*(x^k) = \left[I - \gamma_k \nabla_{22}^2 g(x^k, y^k)\right](y^k - v^{t,k} - y^*(x^k))$$
$$+ \gamma_k \left(\nabla_2 g(x^k, y^*(x^k)) - \nabla_2 g(x^k, y^k) - \nabla_{22}^2 g(x^k, y^k)(y^*(x^k) - y^k)\right).$$

Since $\nabla_{22}^2 g(x, \cdot)$ is $L_{g,2}$-Lipschitz continuous, Lemma 1.2.4 in (Nesterov, 2018) implies that

$$\|\nabla_2 g(x^k, y^*(x^k)) - \nabla_2 g(x^k, y^k) - \nabla_{22}^2 g(x^k, y^k)(y^*(x^k) - y^k)\| \leq \frac{L_{g,2}}{2}\|y^*(x^k) - y^k\|^2,$$

then we can deduce that

$$\|y^k - v^{t+1,k} - y^*(x^k)\| \leq \|I - \gamma_k \nabla_{22}^2 g(x^k, y^k)\|_{\text{op}}\|y^k - v^{t,k} - y^*(x^k)\| + \frac{L_{g,2}\gamma_k}{2}\|y^*(x^k) - y^k\|^2$$

$$\leq (1 - \mu\gamma_k)\|y^k - v^{t,k} - y^*(x^k)\| + \frac{L_{g,2}\gamma_k}{2}\|y^*(x^k) - y^k\|^2, \tag{41}$$

where the second inequality is because $g(x, \cdot)$ is $\mu$-strongly convex and $\gamma_k \leq \frac{1}{L_{g,1}}$. Then we can calculate that

$$\|y^{k+1} - y^*(x^k)\| = \|y^k - v^{T,k} - y^*(x^k)\| \leq (1 - \mu\gamma_k)^T \|y^k - v^{0,k} - y^*(x^k)\| + \frac{L_{g,2}}{2\mu}\|y^*(x^k) - y^k\|^2,$$

$$= (1 - \mu\gamma_k)^T \|y^k - \gamma_k \nabla_2 g(x^k, y^k) - y^*(x^k)\| + \frac{L_{g,2}}{2\mu}\|y^*(x^k) - y^k\|^2$$

$$\leq (1 - \mu\gamma_k)^{T+\frac{1}{2}} \|y^k - y^*(x^k)\| + \frac{L_{g,2}}{2\mu}\|y^*(x^k) - y^k\|^2$$

$$\leq \frac{1}{4}\|y^k - y^*(x^k)\| + \frac{L_{g,2}}{2\mu}\|y^*(x^k) - y^k\|^2, \tag{42}$$

where the second inequality is derived based on the same reason as in (19), and the last ineauqlity holds when $T \geq \frac{\ln 1/4}{\ln(1-\mu\gamma_k)} - \frac{1}{2} = \Theta(\kappa)$. Finally we have

$$\|y^{k+1} - y^*(x^{k+1})\| \leq \|y^{k+1} - y^*(x^k)\| + \|y^*(x^{k+1}) - y^*(x^k)\|$$

$$\overset{(42)}{\leq} \frac{1}{4}\|y^k - y^*(x^k)\| + \frac{L_{g,2}}{2\mu}\|y^*(x^k) - y^k\|^2 + \|y^*(x^{k+1}) - y^*(x^k)\|$$

$$\overset{(33)}{\leq} \frac{1}{4}\|y^k - y^*(x^k)\| + \frac{L_{g,2}}{2\mu}\|y^*(x^k) - y^k\|^2 + \frac{L_{g,1}}{\mu}\|x^{k+1} - x^k\|.$$

Similarily, from the update of $w^{t,k}$ we know that

$$\|u^k - w^{t+1.k} - u^*(x^k, y^k)\| = \|\left[I - \gamma_k \nabla_{22}^2 g(x^k, y^k)\right](u^k - w^{t,k} - u^*(x^k, y^k))\|$$

$$\leq (1 - \mu\gamma_k)\|u^k - w^{t,k} - u^*(x^k, y^k)\|. \tag{43}$$

This inequality implies

$$\begin{aligned}
\|u^{k+1} - u^*(x^k, y^k)\| &= \|u^k - w^{T.k} - u^*(x^k, y^k)\| \leq (1 - \mu\gamma_k)^T \|u^k - w^{0,k} - u^*(x^k, y^k)\| \\
&= (1 - \mu\gamma_k)^T \|u^k - \gamma_k \nabla_{22}^2 g(x^k, y^k) u^k + \gamma_k \nabla_2 f(x^k, y^k) - u^*(x^k, y^k)\| \\
&= (1 - \mu\gamma_k)^T \| \left[ I - \gamma_k \nabla_{22}^2 g(x^k, y^k) \right] (u^k - u^*(x^k, y^k)) \| \\
&\leq (1 - \mu\gamma_k)^{T+1} \|u^k - u^*(x^k)\| + (1 - \mu\gamma_k)^{T+1} \|u^*(x^k, y^k) - u^*(x^k)\| \\
&\leq \frac{1}{4} \|u^k - u^*(x^k)\| + \frac{1}{4} \|u^*(x^k, y^k) - u^*(x^k)\|,
\end{aligned}$$

(44)

where the last inequality holds when $T \geq \frac{\ln 1/4}{\ln(1 - \mu\gamma_k)} - 1$. Then we have

$$\begin{aligned}
\|u^{k+1} - u^*(x^{k+1})\| &\leq \|u^{k+1} - u^*(x^k, y^k)\| + \|u^*(x^k) - u^*(x^k, y^k)\| + \|u^*(x^k) - u^*(x^{k+1})\| \\
&\leq \frac{1}{4} \|u^k - u^*(x^k)\| + \frac{5}{4} \|u^*(x^k, y^k) - u^*(x^k)\| + \|u^*(x^{k+1}) - u^*(x^k)\| \\
&\stackrel{(21),(33)}{\leq} \frac{1}{4} \|u^k - u^*(x^k)\| + \frac{5L_1}{4\mu} \|y^k - y^*(x^k)\| + \frac{L_u}{\mu} \|x^{k+1} - x^k\|.
\end{aligned}$$

Hence we complete the proof. $\qquad\square$

Next, we prove that $\|y^k - y^*(x^k)\|$ and $\|u^k - u^*(x^k)\|$ have uniform upper bounds.

**Lemma E.4.** *Suppose Assumption 3.1 holds. Take $T \geq \Theta(\kappa)$ and*

$$\alpha_k \leq \min\left\{ \frac{\mu^2}{8L_{g,1}L_2L_{g,2}}, \frac{\mu^2}{20L_{g,1}^2 L_1}, \frac{5\mu L_1}{8L_u L_2 L_{g,2}}, \frac{\mu}{4L_{g,1}L_u} \right\} = O(\kappa^{-3}),$$

*we have $\|y^k - y^*(x^k)\| \leq \frac{\mu}{2L_{g,2}}$ and $\|u^k - u^*(x^k)\| \leq \frac{5L_1}{2L_{g,2}}$ for all $0 \leq k \leq K$, then*

$$\|y^{k+1} - y^*(x^{k+1})\| \leq \frac{1}{2} \|y^k - y^*(x^k)\| + \frac{L_{g,1}}{\mu} \|x^{k+1} - x^k\|.$$

(45)

*Proof.* From (26) and $\|u^*(x^k)\| \leq r_u$ we know that

$$\|d_x^k\| = \|\nabla_1 f(x^k, y^k) - \nabla_{12}^2 g(x^k, y^k) u^k\| \leq C_{f,0} + L_{g,1} \|u^k - u^*(x^k)\| + L_{g,1} r_u \leq L_2 + L_{g,1} \|u^k - u^*(x^k)\|. \quad (46)$$

For $\|x^{k+1} - x^k\| = \alpha_k \|d_x^k\|$, combine (46) with (39) and (40), we have

$$\|y^{k+1} - y^*(x^{k+1})\| \leq \frac{L_{g,2}}{2\mu} \|y^*(x^k) - y^k\|^2 + \frac{L_{g,1}\alpha_k}{\mu} \left( L_2 + L_{g,1} \|u^k - u^*(x^k)\| \right) + \frac{1}{4} \|y^k - y^*(x^k)\|,$$

$$\|u^{k+1} - u^*(x^{k+1})\| \leq \frac{1}{4} \|u^k - u^*(x^k)\| + \frac{5L_1}{4\mu} \|y^k - y^*(x^k)\| + \frac{L_u \alpha_k}{\mu} \left( L_2 + L_{g,1} \|u^k - u^*(x^k)\| \right).$$

If $\|y^k - y^*(x^k)\| \leq \frac{\mu}{2L_{g,2}}$, $\|u^k - u^*(x^k)\| \leq \frac{5L_1}{2L_{g,2}}$ and $\alpha_k \leq \min\left\{ \frac{\mu^2}{8L_{g,1}L_2L_{g,2}}, \frac{\mu^2}{20L_{g,1}^2 L_1}, \frac{5\mu L_1}{8L_u L_2 L_{g,2}}, \frac{\mu}{4L_{g,1}L_u} \right\}$, we can deduce that $\|y^{k+1} - y^*(x^{k+1})\| \leq \frac{\mu}{2L_{g,2}}$, $\|u^{k+1} - u^*(x^{k+1})\| \leq \frac{5L_1}{2L_{g,2}}$. By induction, since $\|y^0 - y^*(x^0)\| \leq \frac{\mu}{2L_{g,2}}$, $\|u^0 - u^*(x^0)\| \leq \frac{5L_1}{2L_{g,2}}$ (Lemma D.1), we have $\|y^k - y^*(x^k)\| \leq \frac{\mu}{2L_{g,2}}$, $\|u^k - u^*(x^k)\| \leq \frac{5L_1}{2L_{g,2}}$ for all $0 \leq k \leq K$. Moreover, we can get (45) by taking $\|y^k - y^*(x^k)\| \leq \frac{\mu}{2L_{g,2}}$ in to (39). $\qquad\square$

Then, we can obtain the results in Theorem 2.

**Theorem E.5** (**Restatement of Theorem 3.2**)**.** *Under Assumption 3.1, choose an initial iterate $(y^0, u^0, x^0)$ in BOX 1 that satisfies $\|y^0 - y^*(x^0)\| \leq \min\left\{ \frac{\mu}{2L_{g,2}}, \frac{1}{2\sqrt{L_1}} \right\}$ and $\|u^0 - u^*(x^0)\| \leq \min\left\{ \frac{5L_1}{2L_{g,2}}, \frac{\sqrt{L_1}}{\mu} \right\}$. Then, for any constant step size $\gamma_k = \gamma \leq 1/L_{g,1}$, there exists a proper constant step size*

$$\alpha_k = \alpha \leq \min\left\{ \frac{\mu^2}{8L_{g,1}L_2L_{g,2}}, \frac{5\mu L_1}{8L_u L_2 L_{g,2}}, \frac{\mu}{4L_{g,1}L_u}, \frac{1}{4L_\Phi}, \frac{L_1}{4L_u^2}, \frac{\mu^2}{64L_1 L_{g,1}^2} \right\} = \Theta(\kappa^{-3}),$$

*and $T \geq \Theta(\kappa)$ such that NBO-GD, as described in Algorithm 2, has the following properties:*

(a) *For all integers $K \geq 1$, $\min_{0 \leq k \leq K-1} \|\nabla\Phi(x^k)\|^2 \leq \frac{2\Phi(x^0)-2\Phi^*+4}{\alpha K} = O(\frac{\kappa^3}{K})$. That is, NBO-GD can find an $\epsilon$-optimal solution $\bar{x}$ (i.e., $\|\nabla\Phi(\bar{x})\|^2 \leq \epsilon$) in $K = O(\kappa^3 \epsilon^{-1})$ steps.*

(b) *The computational complexity of NBO-GD is: $O(\kappa^3/\epsilon)$ gradient computations and Jacobian-vector products, and $O(\kappa^4/\epsilon)$ Hessian-vector products.*

*Proof.* Define a Lyapunov function

$$V_k = f(x^k, y^*(x^k)) - \Phi^* + b_y\|y^k - y^*(x^k)\|^2 + b_u\|u^k - u^*(x^k)\|^2,$$

where $b_y = 4L_1, b_u = \frac{\mu^2}{8L_1}$. When $\alpha_k$ satisfies the bound in Lemma E.4, because of Cauchy-Schwarz inequality and $x^{k+1} - x^k = \alpha_k d_x^k$, inequalities (45) and (40) become

$$\|y^{k+1} - y^*(x^{k+1})\|^2 \leq \frac{1}{2}\|y^k - y^*(x^k)\|^2 + \frac{2L_{g,1}^2\alpha_k^2}{\mu^2}\|d_x^k\|^2,$$

$$\|u^{k+1} - u^*(x^{k+1})\|^2 \leq \frac{1}{8}\|u^k - u^*(x^k)\|^2 + \frac{8L_1^2}{\mu^2}\|y^k - y^*(x^k)\|^2 + \frac{4L_u^2\alpha_k^2}{\mu^2}\|d_x^k\|^2.$$

Combining (36), we obtain

$$V_{k+1} - V_k \leq -\frac{\alpha_k}{2}\|\nabla\Phi(x^k)\|^2 - \left(\frac{\alpha_k}{2} - \frac{L_\Phi\alpha_k^2}{2} - \frac{4b_uL_u^2\alpha_k^2}{\mu^2} - \frac{2b_yL_{g,1}^2\alpha_k^2}{\mu^2}\right)\|d_x^k\|^2$$
$$- \left(\frac{b_y}{2} - L_1^2\alpha_k - \frac{8b_uL_1^2}{\mu^2}\right)\|y^k - y^*(x^k)\|^2 - \left(\frac{7b_u}{8} - L_{g,1}^2\alpha_k\right)\|u^k - u^*(x^k)\|^2.$$

Further controlling $\alpha_k \leq \min\left\{\frac{1}{4L_\Phi}, \frac{L_1}{4L_u^2}, \frac{\mu^2}{64L_1L_{g,1}^2}\right\}$, we have

$$\frac{\alpha_k}{2} - \frac{L_\Phi\alpha_k^2}{2} - \frac{4b_uL_u^2\alpha_k^2}{\mu^2} - \frac{2b_yL_{g,1}^2\alpha_k^2}{\mu^2} \geq \frac{\alpha_k}{8}, \quad \frac{b_y}{2} - L_1^2\alpha_k - \frac{8b_uL_1^2}{\mu^2} \geq \frac{b_y}{8}, \quad \frac{7b_u}{8} - L_{g,1}^2\alpha_k \geq \frac{7b_u}{16}.$$

Then, by telescoping sum, we have $\min_{0 \leq k \leq K-1} \|\nabla\Phi(x^k)\|^2 \leq \frac{1}{K}\sum_{k=0}^{K-1}\|\nabla\Phi(x^k)\|^2 \leq \frac{2V_0}{\alpha K} = O\left(\frac{\kappa^3}{K}\right)$, where $V_0 = \Phi(x^0) - \Phi^* + 4L_1\|y^0 - y^*(x^0)\|^2 + \frac{\mu^2}{8L_1}\|u^0 - u^*(x^0)\|^2 \leq \Phi(x^0) - \Phi^* + 2$ because from Lemma D.1 we know that $\|y^0 - y^*(x^0)\|^2 \leq \frac{1}{4L_1}$ and $\|u^0 - u^*(x^0 k)\|^2 \leq \frac{L_1}{\mu^2}$.

Additionally, we calculate the complexities of Algorithm 2 to achieve a stationary point. To achieve $\min_{0 \leq k \leq K-1} \|\nabla\Phi(x^k)\|^2 \leq \epsilon$, we need gradient computations: $\text{Gc}(\epsilon) = O(K + N_0) = O(\kappa^3 \epsilon^{-1})$; Matrix-vector products: $\text{MV}(\epsilon) = O(KT + Q_0) = O(\kappa^4 \epsilon^{-1})$. $\qquad\square$

Moreover, we give a brief proof of Remark 3.4.

*Proof.* **(Proof of Remark 3.4)** For $y^k$ and $v^k$, if we use CG to solve (8) with initial point $v^{-1,k} = 0$, we have

$$\|y^{k+1} - y^*(x^{k+1})\| \leq \|y^{k+1} - y^*(x^k)\| + \|y^*(x^{k+1}) - y^*(x^k)\|$$
$$\overset{(33)}{\leq} \|y^k - v^*(x^k, y^k) - y^*(x^k)\| + \|v^{T,k} - v^*(x^k, y^k)\| + \frac{L_{g,1}}{\mu}\|x^{k+1} - x^k\|$$
$$\leq \frac{L_{g,2}}{2\mu}\|y^k - y^*(x^k)\|^2 + 2\sqrt{\kappa}\left(\frac{\sqrt{\kappa}-1}{\sqrt{\kappa}+1}\right)^{T+1}\|v^{-1,k} - v^*(x^k, y^k)\| + \frac{L_{g,1}}{\mu}\|x^{k+1} - x^k\|$$
$$\leq \frac{L_{g,2}}{2\mu}\|y^k - y^*(x^k)\|^2 + 2\frac{\sqrt{\kappa}L_{g,1}}{\mu}\left(\frac{\sqrt{\kappa}-1}{\sqrt{\kappa}+1}\right)^{T+1}\|y^k - y^*(x^k)\| + \frac{L_{g,1}}{\mu}\|x^{k+1} - x^k\|$$

where the third inequality is due to (6) and the linear rate of CG (refer to (17) in (Grazzi et al., 2020)), and the last inequality folloes from $v^{-1,k} = 0$ and $\|v^*(x^k, y^k)\| = \|[\nabla_{22}^2 g(x^k, y^k)]^{-1}(\nabla_2 g(x^k, y^k) - \nabla_2 g(x^k, y^*(x^k)))\| \leq \frac{L_{g,1}}{\mu}\|y^k - y^*(x^k)\|$.

When we take $T = \Theta(\sqrt{\kappa}\log(\kappa))$, the above inequality becomes the same as (39). For $u^k$ and $w^k$, if we use CG to solve (10) with initial point $w^{-1,k} = 0$, similarily, we obain

$$\|u^{k+1} - u^*(x^k, y^k)\| = \|u^k - w^{T,k} - u^*(x^k, y^k)\|$$

$$\leq \|w^{T,k} - w^*(x^k, y^k, u^k)\| \leq 2\sqrt{\kappa}\big(\frac{\sqrt{\kappa}-1}{\sqrt{\kappa}+1}\big)^{T+1}\|w^{-1,k} - w^*(x^k, y^k, u^k)\|$$

$$= 2\sqrt{\kappa}\big(\frac{\sqrt{\kappa}-1}{\sqrt{\kappa}+1}\big)^{T+1}\|u^k - u^*(x^k, y^k)\|$$

$$\leq 2\sqrt{\kappa}\big(\frac{\sqrt{\kappa}-1}{\sqrt{\kappa}+1}\big)^{T+1}\|u^k - u^*(x^k)\| + 2\sqrt{\kappa}\big(\frac{\sqrt{\kappa}-1}{\sqrt{\kappa}+1}\big)^{T+1}\|u^*(x^k, y^k) - u^*(x^k)\|.$$

When we take $T = \Theta(\sqrt{\kappa}\log(\kappa))$, we can get (44).

The proof process following (39) and (44) is the same as that of NBO-GD. Therefore, if we replace GD with CG in Algorithm 2, the convergence rate is $\min_{0 \leq k \leq K-1} \|\nabla\Phi(x^k)\|^2 \leq \frac{1}{K}\sum_{k=0}^{K-1}\|\nabla\Phi(x^k)\|^2 \leq \frac{2V_0}{\alpha K}$. Additionally, in order to achieve $\min_{0 \leq k \leq K-1}\|\nabla\Phi(x^k)\|^2 \leq \epsilon$, we need gradient computations: $\mathrm{Gc}(\epsilon) = O(K + N_0) = O(\kappa^3\epsilon^{-1})$; Matrix-vector products: $\mathrm{MV}(\epsilon) = O(KT + Q_0) = O(\kappa^{3.5}\log\kappa\epsilon^{-1})$. $\qquad\square$

### E.2. Detailed Proof of Theorem 3.7

Here, we provide a detailed proof of Theorem 3.7 for the stochastic setting. Firstly, similar to the deterministic setting, we establish the descent of $\mathbb{E}[\Phi(x^k)]$, $\mathbb{E}\left[\|y^k - y^*(x^k)\|^2\right]$ and $\mathbb{E}\left[\|u^0 - u^*(x^k)\|^2\right]$.

**Lemma E.6.** *Suppose Assumption 3.1 and 3.5 hold, then the sequences generated by Algorithm 4 satisfy*

$$\mathbb{E}[\Phi(x^{k+1})] \leq \mathbb{E}[\Phi(x^k)] - \frac{\alpha_k}{2}\mathbb{E}\left[\|\nabla\Phi(x^k)\|^2\right] - \big(\frac{\alpha_k}{2} - \frac{L_\Phi\alpha_k^2}{2}\big)\mathbb{E}\left[\|D_x^k\|^2\right] + 2L_1^2\alpha_k\mathbb{E}\left[\|y^k - y^*(x^k)\|^2\right]$$
$$+ \big(2L_{g,1}^2 + 4S_3^{g,2}\big)\alpha_k\mathbb{E}\left[\|u^k - u^*(x^k)\|^2\right] + \big(2S_{f,1} + 4S_3^{g,2}r_u^2\big)\alpha_k. \tag{47}$$

*Proof.* Recall that $\mathbb{E}^k = \mathbb{E}[\cdot|x^k, y^k, u^k]$, from Assumption 3.5 we can deduce that

$$\mathbb{E}^k\left[\|D_x^k - d_x^k\|^2\right]$$
$$\leq 2\mathbb{E}^k\left[\|\nabla_1 F(x^k, y^k; B_3^k) - \nabla_1 f(x^k, y^k)\|^2\right] + 2\mathbb{E}^k\left[\|\nabla_{12}^2 G(x^k, y^k, B_4^k) - \nabla_{12}^2 g(x^k, y^k)\|^2\right]\|u^k\|^2$$
$$\leq 2S_{f,1} + 2S_3^{g,2}\|u^k\|^2 \leq 2S_{f,1} + 2S_3^{g,2}\big(2\|u^k - u^*(x^k)\|^2 + 2\|u^*(x^k)\|^2\big)$$
$$\leq 2S_{f,1} + 4S_3^{g,2}r_u^2 + 4S_3^{g,2}\|u^k - u^*(x^k)\|^2, \tag{48}$$

where the last inequality follows from $\|u^*(x^k)\| \leq r_u$. For $x^{k+1} = x^k - \alpha_k D_x^k$, replace $d_x^k$ with $D_x^k$ in (37) and take expectation, we have

$$\mathbb{E}\left[\Phi(x^{k+1})\right] = \mathbb{E}\left[\Phi(x^k)\right] + \frac{\alpha_k}{2}\mathbb{E}\left[\|\nabla\Phi(x^k) - D_x^k\|^2\right] - \frac{\alpha_k}{2}\mathbb{E}\left[\|\nabla\Phi(x^k)\|^2\right] - \big(\frac{\alpha_k}{2} - \frac{L_\Phi\alpha_k^2}{2}\big)\mathbb{E}\left[\|D_x^k\|^2\right], \tag{49}$$

where

$$\mathbb{E}\left[\|\nabla\Phi(x^k) - D_x^k\|^2\right] \leq 2\mathbb{E}\left[\|\nabla\Phi(x^k) - d_x^k\|^2\right] + 2\mathbb{E}\left[\|D_x^k - d_x^k\|^2\right]$$
$$\overset{(38),(48)}{\leq} 4L_1^2\mathbb{E}\left[\|y^k - y^*(x^k)\|^2\right] + (4L_{g,1}^2 + 8S_3^{g,2})\mathbb{E}\left[\|u^k - u^*(x^k)\|^2\right] + 4S_{f,1} + 8S_3^{g,2}r_u^2. \tag{50}$$

Then we can get the desired inequality by putting (50) into (49). $\qquad\square$

**Lemma E.7.** *Suppose Assumption 3.1,3.5 and 3.6 hold, if $S_1^{g,2} \leq \frac{\mu^2}{4}$, then the sequences generated by Algorithm 4 satisfy*

$$\mathbb{E}\left[\|y^{k+1} - y^*(x^{k+1})\|^2\right] \leq \frac{5rL_{g,2}^2}{\mu^2}\big(\mathbb{E}\left[\|y^*(x^k) - y^k\|^2\right]\big)^2 + r\big[5(1 - \frac{\mu\gamma_k}{2})^{2T+1} + \frac{20S_1^{g,2}}{\mu^2}\big]\mathbb{E}\left[\|y^k - y^*(x^k)\|^2\right]$$
$$+ \frac{5rL_{g,1}^2\alpha_k^2}{\mu^2}\mathbb{E}\left[\|D_x^k\|^2\right] + 5r(1 - \frac{\mu\gamma_k}{2})^{2T}\gamma_k^2 S_{g,1} + \frac{20rS_{g,1}}{\mu^2}. \tag{51}$$

*Proof.* From the iteration of $v^{t,k}$ we know that

$$\|y^k - v^{t+1,k} - y^*(x^k)\| \le \|y^k - v^{t,k} + \gamma_k d_v^k - y^*(x^k)\| + \gamma_k \|D_v^{t,k} - d_v^{t,k}\|$$

$$\overset{(41)}{\le} (1 - \mu\gamma_k)\|y^k - v^{t,k} - y^*(x^k)\| + \frac{L_{g,2}\gamma_k}{2}\|y^*(x^k) - y^k\|^2 + \gamma_k\|D_v^{t,k} - d_v^{t,k}\|. \tag{52}$$

For $\|D_v^{t,k} - d_v^{t,k}\|$, we have

$$\left(\mathbb{E}^{t,k}\left[\|D_v^{t,k} - d_v^{t,k}\|\right]\right)^2 \le \mathbb{E}^{t,k}\left[\|D_v^{t,k} - d_v^{t,k}\|^2\right]$$

$$= \mathbb{E}^{t,k}\left[\|(\nabla_{22}^2 G(x^k, y^k; B_1^{t,k}) - \nabla_{22}^2 g(x^k, y^k))v^{t,k} - \nabla_2 G(x^k, y^k; B_2^k) + \nabla_2 g(x^k, y^k)\|^2\right]$$

$$\le \mathbb{E}^{t,k}\left[\|\nabla_{22}^2 G(x^k, y^k; B_1^{t,k}) - \nabla_{22}^2 g(x^k, y^k)\|^2\right]\|v^{t,k}\|^2 + \|\nabla_2 G(x^k, y^k; B_2^k) - \nabla_2 g(x^k, y^k)\|^2$$

$$\le S_1^{g,2}\|v^{t,k}\|^2 + \|\nabla_2 G(x^k, y^k; B_2^k) - \nabla_2 g(x^k, y^k)\|^2.$$

Then by trangle inequality, we obtain

$$\mathbb{E}^{t,k}\left[\|D_v^{t,k} - d_v^{t,k}\|\right] \le \sqrt{S_1^{g,2}}\|v^{t,k}\| + \|\nabla_2 G(x^k, y^k; B_2^k) - \nabla_2 g(x^k, y^k)\|$$

$$\le \sqrt{S_1^{g,2}}\|y^k - v^{t,k} - y^*(x^k)\| + \sqrt{S_1^{g,2}}\|y^k - y^*(x^k)\| + \|\nabla_2 G(x^k, y^k; B_2^k) - \nabla_2 g(x^k, y^k)\|. \tag{53}$$

Take expectation on both side of the above inequality and combine (52), we have

$$\mathbb{E}\left[\|y^k - v^{t+1,k} - y^*(x^k)\|\right]$$

$$\overset{(52)}{\le} (1 - \mu\gamma_k)\mathbb{E}\left[\|y^k - v^{t,k} - y^*(x^k)\|\right] + \frac{L_{g,2}\gamma_k}{2}\mathbb{E}\left[\|y^*(x^k) - y^k\|^2\right] + \gamma_k\mathbb{E}\left[\|D_v^{t,k} - d_v^{t,k}\|\right]$$

$$\overset{(53)}{\le} (1 - \mu\gamma_k + \gamma_k\sqrt{S_1^{g,2}})\mathbb{E}\left[\|y^k - v^{t,k} - y^*(x^k)\|\right] + \frac{L_{g,2}\gamma_k}{2}\mathbb{E}\left[\|y^*(x^k) - y^k\|^2\right]$$

$$+ \gamma_k\sqrt{S_1^{g,2}}\mathbb{E}\left[\|y^k - y^*(x^k)\|\right] + \gamma_k\mathbb{E}\left[\|\nabla_2 G(x^k, y^k; B_2^k) - \nabla_2 g(x^k, y^k)\|\right].$$

For $S_{g,1} \le \frac{\mu^2}{4}$, we have $1 - \mu\gamma_k + \gamma_k\sqrt{S_1^{g,2}} \le 1 - \frac{\mu\gamma_k}{2}$, this implies

$$\mathbb{E}\left[\|y^{k+1} - y^*(x^{k+1})\|\right] = \mathbb{E}\left[\|y^k - v^{T,k} - y^*(x^k)\|\right] + \mathbb{E}\left[\|y^*(x^{k+1}) - y^*(x^k)\|\right]$$

$$\le (1 - \frac{\mu\gamma_k}{2})^T\mathbb{E}\left[\|y^k - v^{0,k} - y^*(x^k)\|\right] + \frac{L_{g,2}}{\mu}\mathbb{E}\left[\|y^*(x^k) - y^k\|^2\right] + \frac{2\sqrt{S_1^{g,2}}}{\mu}\mathbb{E}\left[\|y^k - y^*(x^k)\|\right]$$

$$+ \mathbb{E}\left[\|y^*(x^{k+1}) - y^*(x^k)\|\right] + \frac{2}{\mu}\mathbb{E}\left[\|\nabla_2 G(x^k, y^k; B_2^k) - \nabla_2 g(x^k, y^k)\|\right],$$

and from (33) we know that $\mathbb{E}\left[\|y^*(x^{k+1}) - y^*(x^k)\|\right] \le \frac{L_{g,1}}{\mu}\mathbb{E}\left[\|x^{k+1} - x^k\|\right]$. Following $(\sum_{i=1}^n a_i)^2 \le n\sum_{i=1}^n a_i^2$ and $(\mathbb{E}[X])^2 \le \mathbb{E}[X^2]$, we can deduce that

$$\left(\mathbb{E}\left[\|y^{k+1} - y^*(x^{k+1})\|\right]\right)^2 \le 5(1 - \frac{\mu\gamma_k}{2})^{2T}\mathbb{E}\left[\|y^k - v^{0,k} - y^*(x^k)\|^2\right] + \frac{5L_{g,2}^2}{\mu^2}\left(\mathbb{E}\left[\|y^*(x^k) - y^k\|^2\right]\right)^2$$

$$+ \frac{5L_{g,1}^2}{\mu^2}\mathbb{E}\left[\|x^{k+1} - x^k\|^2\right] + \frac{20S_1^{g,2}}{\mu^2}\mathbb{E}\left[\|y^k - y^*(x^k)\|^2\right] + \frac{20}{\mu^2}\mathbb{E}\left[\|\nabla_2 G(x^k, y^k; B_2^k) - \nabla_2 g(x^k, y^k)\|^2\right]$$

$$\le 5(1 - \frac{\mu\gamma_k}{2})^{2T+1}\mathbb{E}\left[\|y^k - y^*(x^k)\|^2\right] + \frac{5L_{g,2}^2}{\mu^2}\left(\mathbb{E}\left[\|y^*(x^k) - y^k\|^2\right]\right)^2 + \frac{20S_1^{g,2}}{\mu^2}\mathbb{E}\left[\|y^k - y^*(x^k)\|^2\right]$$

$$+ \frac{5L_{g,1}^2}{\mu^2}\mathbb{E}\left[\|x^{k+1} - x^k\|^2\right] + 5(1 - \frac{\mu\gamma_k}{2})^{2T}\gamma_k^2 S_{g,1} + + \frac{20S_{g,1}}{\mu^2},$$

where the last inequality is because

$$\mathbb{E}\left[\|\nabla_2 G(x^k, y^k; B_2^k) - \nabla_2 g(x^k, y^k)\|^2\right] = \mathbb{E}\left[\mathbb{E}^k\left[\|\nabla_2 G(x^k, y^k; B_2^k) - \nabla_2 g(x^k, y^k)\|^2\right]\right] \le S_{g,1},$$

and

$$\mathbb{E}\left[\|y^k - v^{0,k} - y^*(x^k)\|^2\right] = \mathbb{E}\left[\|y^k - \gamma_k \nabla_2 G(x^k, y^k; B_2^k) - y^*(x^k)\|^2\right]$$

$$=\mathbb{E}\left[\mathbb{E}^k\left[\|y^k - \gamma_k \nabla_2 G(x^k, y^k; B_2^k) - y^*(x^k)\|^2\right]\right] \overset{\text{similar to (25)}}{\leq} (1 - \mu\gamma_k)\mathbb{E}\left[\|y^k - y^*(x^k)\|^2\right] + \gamma_k^2 S_{g,1}.$$

Then we can complete the proof by combining $x^{k+1} - x^k = -\alpha_k D_x^k$ and $\left(\mathbb{E}\left[\|y^{k+1} - y^*(x^{k+1})\|\right]\right)^2 \geq \frac{1}{r}\mathbb{E}\left[\|y^{k+1} - y^*(x^{k+1})\|^2\right]$ from Assumption 3.6. $\square$

**Lemma E.8.** *Suppose Assumption 3.1 and 3.5 hold, if $S_1^{g,2} \leq \frac{\mu^2}{8}$, we have*

$$\mathbb{E}\left[\|u^{k+1} - u^*(x^{k+1})\|^2\right]$$

$$\leq \left(6(1 - \frac{\mu\gamma_k}{2})^{T+2} + \frac{48S_1^{g,1}}{\mu^2} + 12(1 - \frac{\mu\gamma_k}{2})^T S_2^{g,2}\gamma_k^2 + \frac{48S_2^{g,2}}{\mu^2}\right)\mathbb{E}\left[\|u^k - u^*(x^k)\|^2\right]$$

$$+ \frac{L_1^2}{\mu^2}\left(3 + 6(1 - \frac{\mu\gamma_k}{2})^{T+2} + \frac{48S_1^{g,1}}{\mu^2}\right)\mathbb{E}\left[\|y^k - y^*(x^k)\|^2\right] + \frac{3L_u^2}{\mu^2}\mathbb{E}\left[\|x^{k+1} - x^k\|^2\right]$$

$$+ 6(1 - \frac{\mu\gamma_k}{2})^T S_{f,1}\gamma_k^2 + \frac{24S_{f,1}}{\mu^2} + \left(12(1 - \frac{\mu\gamma_k}{2})^T S_2^{g,2}\gamma_k^2 + \frac{48S_2^{g,2}}{\mu^2}\right)r_u^2. \tag{54}$$

*Proof.* The iteration of $w^{t,k}$ implies that

$$\|u^k - w^{t+1,k} - u^*(x^k, y^k)\|^2 = (1 + \mu\gamma_k)\|u^k - u^{t,k} + \gamma_k d_w^k - u^*(x^k, y^k)\|^2 + (1 + \frac{1}{\mu\gamma_k})\gamma_k^2\|D_w^{t,k} - d_w^{t,k}\|^2$$

$$\overset{(43)}{\leq} (1 + \mu\gamma_k)(1 - \mu\gamma_k)^2\|u^k - w^{t,k} - u^*(x^k, y^k)\|^2 + \frac{2\gamma_k}{\mu}\|D_w^{t,k} - d_w^{t,k}\|^2. \tag{55}$$

For $\|D_w^{t,k} - d_w^{t,k}\|$ we have

$$\mathbb{E}^k\left[\|D_w^{t,k} - d_w^{t,k}\|^2\right] = \mathbb{E}^k\left[\mathbb{E}^{t,k}\left[\|D_w^{t,k} - d_w^{t,k}\|^2\right]\right]$$

$$\leq \mathbb{E}^k\left[\mathbb{E}^{t,k}\left[\|(\nabla_{22}^2 G(x^k, y^k; B_1^{t,k}) - \nabla_{22}^2 g(x^k, y^k))\|^2\right]\|w^{t,k}\|^2\right] + \mathbb{E}^k\left[\| - D_u^k + d_u^k\|^2\right]$$

$$\leq S_1^{g,1}\mathbb{E}^k\left[\|w^{t,k}\|^2\right] + \mathbb{E}^k\left[\|D_u^k - d_u^k\|^2\right]$$

$$\leq S_1^{g,1}\mathbb{E}^k\left[\|w^{t,k}\|^2\right] + \mathbb{E}^k\left[\|(\nabla_{22}^2 G(x^k, y^k; B_1^k) - \nabla_{22}^2 g(x^k, y^k))\|u^k\|^2 - (\nabla_2 F(x^k, y^k; B_3^k) - \nabla_2 f(x^k, y^k))\|^2\right]$$

$$\leq S_1^{g,1}\mathbb{E}^k\left[\|w^{t,k}\|^2\right] + 2S_2^{g,2}\|u^k\|^2 + 2S_{f,1}$$

$$\leq 2S_1^{g,1}\mathbb{E}^k\left[\|u^k - w^{t,k} - u^*(x^k, y^k)\|^2\right] + 2S_1^{g,1}\|u^k - u^*(x^k, y^k)\|^2 + 2S_2^{g,2}\|u^k\|^2 + 2S_{f,1}.$$

Take expectation and combine (55),

$$\mathbb{E}\left[\|u^k - w^{t+1,k} - u^*(x^k, y^k)\|^2\right]$$

$$\leq (1 - \mu\gamma_k + \frac{4S_1^{g,1}\gamma_k}{\mu})\mathbb{E}\left[\|u^k - w^{t,k} - u^*(x^k, y^k)\|^2\right] + \frac{4S_1^{g,1}\gamma_k}{\mu}\mathbb{E}\left[\|u^k - u^*(x^k, y^k)\|^2\right]$$

$$+ \frac{4S_2^{g,2}\gamma_k}{\mu}\mathbb{E}\left[\|u^k\|^2\right] + \frac{4\gamma_k S_{f,1}}{\mu}.$$

Since $S_{g,1} \leq \frac{\mu^2}{8}$, we obtain

$$\mathbb{E}\left[\|u^{k+1} - u^*(x^k, y^k)\|^2\right] = \mathbb{E}\left[\|u^k - w^{T,k} - u^*(x^k, y^k)\|^2\right]$$

$$\leq (1 - \frac{\mu\gamma_k}{2})^T\mathbb{E}\left[\|u^k - w^{0,k} - u^*(x^k, y^k)\|^2\right] + \frac{8S_1^{g,1}}{\mu^2}\mathbb{E}\left[\|u^k - u^*(x^k, y^k)\|^2\right] + \frac{8S_2^{g,2}}{\mu^2}\mathbb{E}\left[\|u^k\|^2\right] + \frac{8S_{f,1}}{\mu^2}$$

$$\leq \left((1 - \frac{\mu\gamma_k}{2})^{T+2} + \frac{8S_1^{g,1}}{\mu^2}\right)\mathbb{E}\left[\|u^k - u^*(x^k, y^k)\|^2\right] + \left(2(1 - \frac{\mu\gamma_k}{2})^T S_2^{g,2}\gamma_k^2 + \frac{8S_2^{g,2}}{\mu^2}\right)\mathbb{E}\left[\|u^k\|^2\right]$$

$$+ 2(1 - \frac{\mu\gamma_k}{2})^T S_{f,1}\gamma_k^2 + \frac{8S_{f,1}}{\mu^2}, \tag{56}$$

where the last inequlity is becuase

$$\mathbb{E}\left[\|u^k - w^{0,k} - u^*(x^k, y^k)\|^2\right] = \mathbb{E}\left[\mathbb{E}^k\left[\|u^k - \gamma_k D_u^k - u^*(x^k, y^k)\|^2\right]\right]$$

$$\leq \mathbb{E}\left[\|u^k - \gamma_k d_u^k - u^*(x^k, y^k)\|^2\right] + \gamma_k^2 \mathbb{E}\left[\mathbb{E}^k\left[\|D_u^k - d_u^k\|^2\right]\right]$$

$$\leq \mathbb{E}\left[\|[I - \gamma_k \nabla_{22}^2 g(x^k, y^k)](u^k - u^*(x^k, y^k))\|^2\right] + 2S_2^{g,2}\gamma_k^2 \mathbb{E}\left[\|u^k\|^2\right] + 2S_{f,1}\gamma_k^2$$

$$\leq (1 - \mu\gamma_k)^2 \mathbb{E}\left[\|u^k - u^*(x^k, y^k)\|^2\right] + 2S_2^{g,2}\gamma_k^2 \mathbb{E}\left[\|u^k\|^2\right] + 2S_{f,1}\gamma_k^2.$$

Then by $(\sum_{i=1}^n a_i)^2 \leq n \sum_{i=1}^n a_i^2$, we have

$$\mathbb{E}\left[\|u^{k+1} - u^*(x^{k+1})\|^2\right]$$

$$\leq 3\mathbb{E}\left[\|u^{k+1} - u^*(x^k, y^k)\|^2\right] + 3\mathbb{E}\left[\|u^*(x^k, y^k) - u^*(x^k)\|^2\right] + 3\mathbb{E}\left[\|u^*(x^{k+1}) - u^*(x^k)\|^2\right]$$

$$\overset{(56),(21),(33)}{\leq} \left(3(1 - \frac{\mu\gamma_k}{2})^{T+2} + \frac{24S_1^{g,1}}{\mu^2}\right)\mathbb{E}\left[\|u^k - u^*(x^k, y^k)\|^2\right] + \left(6(1 - \frac{\mu\gamma_k}{2})^T S_2^{g,2}\gamma_k^2 + \frac{24S_2^{g,2}}{\mu^2}\right)\mathbb{E}\left[\|u^k\|^2\right]$$

$$+ \frac{3L_1^2}{\mu^2}\mathbb{E}\left[\|y^k - y^*(x^k)\|^2\right] + \frac{3L_u^2}{\mu^2}\mathbb{E}\left[\|x^{k+1} - x^k\|^2\right] + 6(1 - \frac{\mu\gamma_k}{2})^T S_{f,1}\gamma_k^2 + \frac{24S_{f,1}}{\mu^2}$$

$$\leq \left(6(1 - \frac{\mu\gamma_k}{2})^{T+2} + \frac{48S_1^{g,1}}{\mu^2} + 12(1 - \frac{\mu\gamma_k}{2})^T S_2^{g,2}\gamma_k^2 + \frac{48S_2^{g,2}}{\mu^2}\right)\mathbb{E}\left[\|u^k - u^*(x^k)\|^2\right]$$

$$+ \frac{L_1^2}{\mu^2}\left(3 + 6(1 - \frac{\mu\gamma_k}{2})^{T+2} + \frac{48S_1^{g,1}}{\mu^2}\right)\mathbb{E}\left[\|y^k - y^*(x^k)\|^2\right] + \frac{3L_u^2\alpha_k^2}{\mu^2}\mathbb{E}\left[\|D_x^k\|^2\right]$$

$$+ 6(1 - \frac{\mu\gamma_k}{2})^T S_{f,1}\gamma_k^2 + \frac{24S_{f,1}}{\mu^2} + \left(12(1 - \frac{\mu\gamma_k}{2})^T S_2^{g,2}\gamma_k^2 + \frac{48S_2^{g,2}}{\mu^2}\right)r_u^2,$$

where the last inequality follows from

$$\mathbb{E}\left[\|u^k - u^*(x^k, y^k)\|^2\right] \leq 2\mathbb{E}\left[\|u^k - u^*(x^k)\|^2\right] + 2\mathbb{E}\left[\|u^*(x^k, y^k) - u^*(x^k)\|^2\right]$$

$$\leq 2\mathbb{E}\left[\|u^k - u^*(x^k)\|^2\right] + \frac{2L_1^2}{\mu^2}\mathbb{E}\left[\|y^k - y^*(x^k)\|^2\right],$$

and $\mathbb{E}\left[\|u^k\|^2\right] \leq 2\mathbb{E}\left[\|u^k - u^*(x^k)\|^2\right] + 2r_u^2$. $\qquad\square$

Next, we prove that there exist constant bounds for both $\mathbb{E}\left[\|y^k - y^*(x^k)\|^2\right]$ and $\mathbb{E}\left[\|u^k - u^*(x^k)\|^2\right]$.

**Lemma E.9.** *Suppose Assumptions 3.1, 3.5 and 3.6 hold, choose $T$, stepsize $\alpha_k$ and batch sizes satisfy the conditions*

$$T \geq \frac{\max\{\ln(1/\sqrt{40r}), \ln(1/96)\}}{\ln(1 - \mu\gamma_k/2)} = \Theta(\kappa),$$

$$\alpha_k \leq \bar{\alpha} := \min\left\{\frac{\mu}{6\sqrt{2}L_u L_{g,1}}, \frac{\mu^2}{8\sqrt{30r}L_1 L_{g,1}^2}, \frac{\mu^2}{80r L_{g,1} L_2 L_{g,2}}, \frac{L_1\mu}{6\sqrt{10r}L_{g,2}L_u L_2}\right\} = \Theta(\kappa^{-3}),$$

$$S_{f,1} \leq \min\left\{\frac{L_1^2\mu^2}{375r L_{g,2}^2}, L_2^2\right\}, \; S_{g,1} \leq \frac{\mu^4}{3360r^2 L_{g,2}^2},$$

$$S_1^{g,1} \leq \min\left\{\frac{\mu^2}{160r}, \frac{\mu^2}{768}\right\}, \; S_2^{g,2} \leq \min\left\{\frac{L_1^2\mu^2}{735r L_{g,2}^2 r_u^2}, \frac{\mu^2}{768}\right\}, \; S_3^{g,2} \leq \min\left\{L_{g,1}^2, \frac{L_2^2}{r_u^2}\right\},$$

*then we have $\mathbb{E}\left[\|y^k - y^*(x^k)\|^2\right] \leq \frac{\mu^2}{20r L_{g,2}^2}$, $\mathbb{E}\left[\|u^k - u^*(x^k)\|^2\right] \leq \frac{4L_1^2}{5r L_{g,2}^2}$ for all $0 \leq k \leq K$.*

*Proof.* Take $S_3^{g,2} \leq \min\left\{L_{g,1}^2, \frac{L_2^2}{r_u^2}\right\}$, $S_{f,1} \leq L_2^2$, from (46) and (48) we know that

$$\mathbb{E}\left[\|D_x^k\|^2\right] \leq \mathbb{E}\left[\|d_x^k\|^2\right] + \mathbb{E}\left[\mathbb{E}^k\left[\|D_x^k - d_x^k\|^2\right]\right] \leq \left(2L_{g,1}^2 + 4S_3^{g,2}\right)\|u^k - u^*(x^k)\|^2 + 2L_2^2 + 2S_{f,1} + 4S_3^{g,2}r_u^2$$

$$\leq 6L_{g,1}^2\|u^k - u^*(x^k)\|^2 + 8L_2^2. \tag{57}$$

Take $T \geq \frac{\max\{\ln(1/\sqrt{40r}), \ln(1/96)\}}{\ln(1 - \mu\gamma_k/2)}$, $S_1^{g,1} \leq \min\left\{\frac{\mu^2}{160r}, \frac{\mu^2}{768}\right\}$, $S_2^{g,2} \leq \frac{\mu^2}{768}$, combine $\gamma_k \leq \frac{1}{L_{f,1}}$ and $r \geq 1$, then (51) and (54) becomes

$$\mathbb{E}\left[\|y^{k+1} - y^*(x^{k+1})\|^2\right]$$
$$\leq \frac{5rL_{g,2}^2}{\mu^2}\left(\mathbb{E}\left[\|y^*(x^k) - y^k\|^2\right]\right)^2 + \frac{1}{4}\mathbb{E}\left[\|y^k - y^*(x^k)\|^2\right] + \frac{5rL_{g,1}^2\alpha_k^2}{\mu^2}\mathbb{E}\left[\|D_x^k\|^2\right] + \frac{21rS_{g,1}}{\mu^2}, \tag{58}$$

and

$$\mathbb{E}\left[\|u^{k+1} - u^*(x^{k+1})\|^2\right]$$
$$\leq \frac{1}{4}\mathbb{E}\left[\|u^k - u^*(x^k)\|^2\right] + \frac{4L_1^2}{\mu^2}\mathbb{E}\left[\|y^k - y^*(x^k)\|^2\right] + \frac{3L_u^2\alpha_k^2}{\mu^2}\mathbb{E}\left[\|D_x^k\|^2\right] + \frac{25S_{f,1}}{\mu^2} + \frac{49S_2^{g,2}}{\mu^2}r_u^2. \tag{59}$$

Put (57) into the above inequalities, we obtain

$$\mathbb{E}\left[\|y^{k+1} - y^*(x^{k+1})\|^2\right] \leq \frac{5rL_{g,2}^2}{\mu^2}\left(\mathbb{E}\left[\|y^*(x^k) - y^k\|^2\right]\right)^2$$
$$+ \frac{1}{4}\mathbb{E}\left[\|y^k - y^*(x^k)\|^2\right] + \frac{30rL_{g,1}^4\alpha_k^2}{\mu^2}\mathbb{E}\left[\|u^k - u^*(x^k)\|^2\right] + \frac{40rL_{g,1}^2L_2^2\alpha_k^2}{\mu^2} + \frac{21rS_{g,1}}{\mu^2}, \tag{60}$$

and

$$\mathbb{E}\left[\|u^{k+1} - u^*(x^{k+1})\|^2\right]$$
$$\leq \left(\frac{1}{4} + \frac{18L_{g,1}^2L_u^2\alpha_k^2}{\mu^2}\right)\mathbb{E}\left[\|u^k - u^*(x^k)\|^2\right] + \frac{4L_1^2}{\mu^2}\mathbb{E}\left[\|y^k - y^*(x^k)\|^2\right] + \frac{24L_u^2L_2^2\alpha_k^2}{\mu^2} + \frac{25S_{f,1}}{\mu^2} + \frac{49S_2^{g,2}}{\mu^2}r_u^2. \tag{61}$$

For $y^0, u^0$ outputed by BOX 2 satisfy $\mathbb{E}\left[\|y^0 - y^*(x^0)\|^2\right] \leq \frac{\mu^2}{20rL_{g,2}^2}$ and $\mathbb{E}\left[\|u^0 - u^*(x^0)\|^2\right] \leq \frac{4L_1^2}{5rL_{g,2}^2}$, then we complete the proof by induction: If $\mathbb{E}\left[\|y^k - y^*(x^k)\|^2\right] \leq \frac{\mu^2}{20rL_{g,2}^2}$, $\mathbb{E}\left[\|u^k - u^*(x^k)\|^2\right] \leq \frac{4L_1^2}{5rL_{g,2}^2}$, and

$$\alpha_k \leq \min\left\{\frac{\mu}{6\sqrt{2}L_uL_{g,1}}, \frac{\mu^2}{8\sqrt{30r}L_1L_{g,1}^2}, \frac{\mu^2}{80rL_{g,1}L_2L_{g,2}}, \frac{L_1\mu}{6\sqrt{10r}L_{g,2}L_uL_2}\right\}, \tag{62}$$

$$S_{f,1} \leq \frac{L_1^2\mu^2}{375rL_{g,2}^2}, \quad S_{g,1} \leq \frac{\mu^4}{3360r^2L_{g,2}^2}, \quad S_2^{g,2} \leq \frac{L_1^2\mu^2}{735rL_{g,2}^2r_u^2}, \tag{63}$$

we can deduce that $\mathbb{E}\left[\|y^{k+1} - y^*(x^{k+1})\|^2\right] \leq \frac{\mu^2}{20rL_{g,2}^2}$ and $\mathbb{E}\left[\|u^{k+1} - u^*(x^{k+1})\|^2\right] \leq \frac{4L_1^2}{5rL_{g,2}^2}$ from (60) and (61). $\qquad\square$

Finally, we can prove Theorem 3.7.

**Theorem E.10** (**Restatement of Theorem 3.7**). *Under Assumptions 3.1, 3.5 and 3.6, choose an initial iterate $(y^0, u^0, x^0)$ in BOX 2 that satisfies $\mathbb{E}\left[\|y^0 - y^*(x^0)\|^2\right] \leq \min\left\{\frac{\mu^2}{20rL_{g,2}^2}, \frac{1}{4L_1}\right\}$ and $\mathbb{E}\left[\|u^0 - u^*(x^0)\|^2\right] \leq \min\left\{\frac{4L_1^2}{5rL_{g,2}^2}, \frac{L_1}{\mu^2}\right\}$. Then, for any constant step size $\gamma_k = \gamma \leq 1/L_{g,1}$, there exists a proper constant step size $\alpha_k = \alpha = \Theta(\kappa^{-3})$ and $T \geq \Theta(\kappa)$ such that NSBO-SGD, as described in Algorithm 4, has the following properties:*

(a) *Fix $K \geq 1$. For samples with batch sizes $|B_1^{t,k}| \geq \Theta(\kappa^2)$, $|B_1^k| \geq \Theta(\kappa K + \kappa^2)$, $|B_2^k| \geq \Theta(\kappa^3 K + \kappa^4)$, $|B_3^k| \geq \Theta(\kappa^{-1}K)$, $|B_4^k| \geq \Theta(\kappa^{-1}K)$, it holds that $\min_{0 \leq k \leq K-1} \mathbb{E}\left[\|\nabla\Phi(x^k)\|^2\right] = O(\frac{\kappa^3}{K})$. That is, NSBO-SGD can find an $\epsilon$-optimal solution in $K = O(\kappa^3\epsilon^{-1})$ steps.*

(b) *The computational complexity of NSBO-SGD is: $O(\kappa^5\epsilon^{-2})$ gradient complexity for F, $O(\kappa^9\epsilon^{-2})$ gradient complexity for G, $O(\kappa^5\epsilon^{-2})$ Jacobian-vector product complexity, $O(\kappa^7\epsilon^{-2})$ Hessian-vector product complexity.*

*Proof.* Define a Lyapunov function

$$V_k = \mathbb{E}\left[f(x^k, y^*(x^k))\right] - \Phi^* + b_y\mathbb{E}\left[\|y^k - y^*(x^k)\|^2\right] + b_u\mathbb{E}\left[\|u^k - u^*(x^k)\|^2\right],$$

where $b_y = 4L_1, b_u = \frac{\mu^2}{8L_1}$. If the step sizes and $T$ satisfy the conditions in Lemma E.9, for $\mathbb{E}\left[\|y^k - y^*(x^k)\|^2\right] \leq \frac{\mu^2}{20rL_{g,2}^2}$, the inequality (58) becomes

$$\mathbb{E}\left[\|y^{k+1} - y^*(x^{k+1})\|^2\right] \leq \frac{1}{2}\mathbb{E}\left[\|y^k - y^*(x^k)\|^2\right] + \frac{5rL_{g,1}^2\alpha_k^2}{\mu^2}\mathbb{E}\left[\|D_x^k\|^2\right] + \frac{21rS_{g,1}}{\mu^2}, \tag{64}$$

Combining (47) and (59), we have

$$
\begin{aligned}
V_{k+1} - V_k \leq &-\frac{\alpha_k}{2}\mathbb{E}\left[\|\nabla\Phi(x^k)\|^2\right] - \left(\frac{\alpha_k}{2} - \frac{L_\Phi\alpha_k^2}{2} - \frac{5rb_yL_{g,1}^2\alpha_k^2}{\mu^2} - \frac{3b_uL_u^2\alpha_k^2}{\mu^2}\right)\mathbb{E}\left[\|D_x^k\|^2\right] \\
&- \left(\frac{b_y}{2} - 2L_1^2\alpha_k - \frac{4b_uL_1^2}{\mu^2}\right)\mathbb{E}\left[\|y^k - y^*(x^k)\|^2\right] - \left(\frac{3b_u}{4} - 2L_{g,1}^2\alpha_k - 4S_3^{g,2}\alpha_k\right)\mathbb{E}\left[\|u^k - u^*(x^k)\|^2\right] \\
&+ \left(2S_{f,1} + 4S_3^{g,2}r_u^2\right)\alpha_k + \frac{21rb_yS_{g,1}}{\mu^2} + \frac{25b_uS_{f,1}}{\mu^2} + \frac{49b_uS_2^{g,2}r_u^2}{\mu^2}.
\end{aligned} \tag{65}
$$

On the basis of conditions in Lemma E.9, further add

$$\alpha_k \leq \min\left\{\frac{1}{4L_\Phi}, \frac{\mu^2}{160rL_1L_{g,1}^2}, \frac{L_1}{3L_u^2}\right\} = \Theta(\kappa^{-3}), \quad S_3^{g,2} \leq \frac{\mu^2}{128L_1\alpha_k}, \tag{66}$$

then we can deduce that coefficients in (65) are all positive:

$$\frac{\alpha_k}{2} - \frac{L_\Phi\alpha_k^2}{2} - \frac{5rb_yL_{g,1}^2\alpha_k^2}{\mu^2} - \frac{3b_uL_u^2\alpha_k^2}{\mu^2} \geq \frac{\alpha_k}{8}, \quad \frac{b_y}{2} - 2L_1^2\alpha_k - \frac{4b_uL_1^2}{\mu^2} \geq \frac{b_y}{4}, \quad \frac{3b_u}{4} - 2L_{g,1}^2\alpha_k - 4S_3^{g,2}\alpha_k \geq \frac{b_u}{4}.$$

Take $\alpha_k$ as constant step sizes $\alpha$, we have

$$\frac{\alpha}{2}\mathbb{E}\left[\|\nabla\Phi(x^k)\|^2\right] \leq V_k - V_{k+1} + \left(2S_{f,1} + 4S_3^{g,2}r_u^2\right)\alpha + \frac{21rb_yS_{g,1}}{\mu^2} + \frac{25b_uS_{f,1}}{\mu^2} + \frac{49b_uS_2^{g,2}r_u^2}{\mu^2}.$$

By telescoping we obtain

$$
\begin{aligned}
\min_{0\leq k\leq K-1}\mathbb{E}\left[\|\nabla\Phi(x^k)\|^2\right] &\leq \frac{1}{K}\sum_{k=0}^{K-1}\mathbb{E}\left[\|\nabla\Phi(x^k)\|^2\right] \\
&\leq \frac{2V_0}{\alpha K} + 4S_{f,1} + 8S_3^{g,2}r_u^2 + \frac{42rb_yS_{g,1}}{\mu^2\alpha} + \frac{50b_uS_{f,1}}{\mu^2\alpha} + \frac{98b_uS_2^{g,2}r_u^2}{\mu^2\alpha},
\end{aligned}
$$

where $V_0 \leq \mathbb{E}\left[f(x^0, y^*(x^0))\right] - \Phi^* + 2$ because of Lemma D.2. Take $S_{g,1} = O(\frac{\mu^3}{LK}), S_{f,1} = O(\frac{L}{\mu K}), S_2^{g,2} = O(\frac{\mu L}{K}), S_3^{g,2} = O(\frac{L^3}{\mu K})$, we have $\min_{0\leq k\leq K-1}\mathbb{E}\left[\|\nabla\Phi(x^k)\|^2\right] = O(\frac{\kappa^3}{K})$. Combining the conditions in Lemma E.9, we can deduce the batch sizes from (32) as follows $|B_1^{t,k}| \geq \Theta(\kappa^2), |B_1^k| \geq \Theta(\kappa K + \kappa^2), |B_2^k| \geq \Theta(\kappa^3 K + \kappa^4), |B_3^k| \geq \Theta(\kappa^{-1}K), |B_4^k| \geq \Theta(\kappa^{-1}K)$.

Moreover, to achieve $\min_{0\leq k\leq K-1}\mathbb{E}\left[\|\nabla\Phi(x^k)\|^2\right] \leq \epsilon$, we need to choose $K \geq \Theta(\frac{\kappa^3}{\epsilon})$, and $|B_1^{t,k}| \geq \Theta(\kappa^2), |B_1^k| \geq \Theta(\frac{\kappa^4}{\epsilon}), |B_2^k| \geq \Theta(\frac{\kappa^6}{\epsilon}), |B_3^k| \geq \Theta(\frac{\kappa^2}{\epsilon}), |B_4^k| \geq \Theta(\frac{\kappa^2}{\epsilon})$. Then, we need the following complexities: Gradient complexity of $F$: $K|B_3^k| + Q_0|B_0'| = O(\kappa^5\epsilon^{-2})$; Gradient complexity of $G$: $K|B_2^k| + T_0|B_0| = O(\kappa^9\epsilon^{-2})$; Jacobian-vector product complexity: $K|B_4^k| = O(\kappa^5\epsilon^{-2})$; Hessian-vector product complexity: $KT|B_1^{t,k}| + K|B_1^k| + Q_0|B_0| = O(\kappa^7\epsilon^{-2})$. $\quad\square$

