# OpenReview forum: "Efficient Curvature-Aware Hypergradient Approximation for Bilevel Optimization"
_ICML.cc/2025/Conference — ICML 2025 poster_

### Official Review · Reviewer_ihhr · 2025-03-10

**Overall Recommendation:** 3

**Summary:**

The paper focus on bilevel optimization and incorporates curvature information into the approximation of hypergradients in bilevel optimization.  The authors propose a Newton-based framework (NBO) that solves lower-level problems with computing Hessian inverse-vector products.  They establish the convergence rate guarantees in both deterministic and stochastic scenarios, demonstrating improved computational complexity over popular gradient-based methods.  Numerical experiments validate the effectiveness of the proposed method.

**Claims And Evidence:**

Yes

**Essential References Not Discussed:**

The key contribution is in quadratic programming-based bilevel methods; however, the literature on this topic is incomplete, e.g. [R1]-[R3]. Most notably, [R3], which employs a constant batch size for the stochastic setting, in contrast to the increasing batch size required by the proposed method.

[R1] Non-Convex Bilevel Games with Critical Point Selection Maps. Michael Arbel, et. al. NeurIPS 2022.

[R2] A Generalized Alternating Method for Bilevel Optimization under the Polyak-Łojasiewicz Condition. Quan Xiao, et. al. NeurIPS 2023.

[R3] Single-Timescale Multi-Sequence Stochastic Approximation Without Fixed Point Smoothness: Theories and Applications. Yue Huang, et. al.

Furthermore, since their method leverages second-order information, it is essential to compare it with fully first-order bilevel methods to demonstrate that the use of second-order terms does not impede numerical performance. However, the paper lacks references to the relevant literature in this area.

[R4] On Penalty-based Bilevel Gradient Descent Method. Han Shen, et. al. ICML 2023.

**Experimental Designs Or Analyses:**

Yes

**Methods And Evaluation Criteria:**

Yes

**Other Comments Or Suggestions:**

No

**Other Strengths And Weaknesses:**

Strengths:

1. The paper presents a novel method for hypergradient approximation that efficiently incorporates curvature information with rigorous convergence rate guarantees and improves rate in deterministic setting.
2. Numerical experiments showcase the effectiveness of the proposed method, showing improved performance over existing gradient-based methods.

Weaknesses:

1. The theoretical improvements are established only in the deterministic setting. Moreover, the analysis for the stochastic setting requires an increasing batch size inversely proportional to $\epsilon$, while other bilevel methods use a constant batch size (see [R3], [R5]–[R7]). This requirement is impractical for large-scale machine learning applications. Also the lower-level strongly convexity is somewhat restrictive.

2. Although the paper primarily builds upon a fully single-loop second order method, it would be beneficial to include a more comprehensive convergence rate comparison with other bilevel approaches, such as fully first-order methods in discussion and Table 1.

[R1] Non-Convex Bilevel Games with Critical Point Selection Maps. Michael Arbel, et. al. NeurIPS 2022.

[R2] A Generalized Alternating Method for Bilevel Optimization under the Polyak-Łojasiewicz Condition. Quan Xiao, et. al. NeurIPS 2023.

[R3] Single-Timescale Multi-Sequence Stochastic Approximation Without Fixed Point Smoothness: Theories and Applications. Yue Huang, et. al.

[R4] On Penalty-based Bilevel Gradient Descent Method. Han Shen, et. al. ICML 2023.

[R5] A framework for bilevel optimization that enables stochastic and global variance reduction algorithms. Mathieu Dagréou, et. al. NeurIPS 2022.

[R6] Closing the Gap: Tighter Analysis of Alternating Stochastic Gradient Methods for Bilevel Problems. Tianyi Chen, et. al. NeurIPS 2021.

[R7] A Fully Single Loop Algorithm for Bilevel Optimization without Hessian Inverse. Junyi Li, et. al. AAAI 2022.

**Questions For Authors:**

See weakness

**Relation To Broader Scientific Literature:**

This paper achieves enhanced iteration complexity for bilevel algorithm with strongly convex lower-level problem in terms of the condition number in determinstic setting.

**Theoretical Claims:**

I did not fully check the correctness of the proofs, but the theoretical claims seems reasonable.

---

> ### Author Rebuttal · Authors · 2025-04-01
>
> Thank you for your valuable feedback. We will address each point in detail. The figures and the tables mentioned below are in this anonymous link: https://drive.google.com/file/d/1lCY1UF3isNnoujM8AGCPIdJlRqHctj-b/view?usp=sharing
>
> ## Essential References Not Discussed:
>
> * **Comparison with fully first-order methods (numerical evaluation):** **We clarify that we have compared our method with the fully first-order method F2SA [1] in Experiments (Section 4.2 and 4.3).**  Our method consistently outperforms F2SA, even when using small batch sizes (64 or 256). It is worth noting that, based on the results from [2,3], F2SA is considered a SOTA fully first-order bilevel algorithm.
>
>   Furthermore, we conducted **additional experiments on meta-learning** using the Omniglot and miniImageNet (approximately 3GB) datasets, with 4-layer convolutional neural networks (CNN4). We compared our NBO with the SOTA algorithms PZOBO [4] and qNBO [5]. The results, presented in Fig. 1 and 2 of the [anonymous link](https://drive.google.com/file/d/1lCY1UF3isNnoujM8AGCPIdJlRqHctj-b/view?usp=sharing), show that NBO consistently outperforms the other methods on both datasets, highlighting the effectiveness of our framework.
>
>   Note that PZOBO is a widely used Hessian-free bilevel algorithm, while qNBO is a recently proposed curvature-aware algorithm that employs the quasi-Newton method to solve the lower-level problem.
>
> * **More references:** First, among the references mentioned by the reviewer, [R1, R2, R4] focus on nonconvex lower-level problems, whereas our work focuses on strongly convex lower-level problems. Second, the reference [R3] does not propose a new bilevel algorithm but rather provides an improved analysis for SOBA. Since SOBA is a special case of our NBO framework (as noted in Remark 2.1 (i) of our work), their analysis also applies to our NSBO-SGD when $T=0$.
>
>   We appreciate the reviewer’s suggestions and will include these references in our work. Additionally, we have added a summary table for the stochastic setting that incorporates the references mentioned by the reviewer, as shown in Table 1 in the [anonymous link](https://drive.google.com/file/d/1lCY1UF3isNnoujM8AGCPIdJlRqHctj-b/view?usp=sharing) (the second page).
>
> ## Other Strengths And Weaknesses:
>
> * **In the stochastic setting, we also establish a theoretical improvement:** As stated after Theorem 3.7, our NSBO-SGD improves upon the SOTA result of AmIGO by a factor of $\log \kappa$ in the stochastic setting, where AmIGO also employs a large batch size.
> * **Concern about batch size:** The large batch size we choose helps achieve better sample complexity. If the batch size is set to $O(1)$, we can obtain a sample complexity of $O(\kappa^{16} \epsilon^{-2})$, which is worse than $O(\kappa^9 \epsilon^{-2})$ in Theorem 3.7. It is also worth noting that SOBA is a special case of our NBO framework.
> * **Extension to non-strongly convex problems:** The NBO framework is primarily designed for bilevel optimization problems with a strongly convex structure. For non-strongly convex problems, existing methods often involve reformulating the original problem to incorporate a strongly convex structure. Once this structure is established, our NBO framework can be applied. For instance, we compared the BAMM method [7] and BAMM+NBO (i.e., using NBO to compute $d_x^k$ in BAMM). The result, presented in Fig. 3 of the [anonymous link](https://drive.google.com/file/d/1lCY1UF3isNnoujM8AGCPIdJlRqHctj-b/view?usp=sharing), shows that BAMM+NBO significantly outperforms BAMM, highlighting the effectiveness of the NBO framework.
> * **Comparison with fully first-order methods (convergence rate and complexity):** Please see Table 1 in the [anonymous link](https://drive.google.com/file/d/1lCY1UF3isNnoujM8AGCPIdJlRqHctj-b/view?usp=sharing) for the stochastic setting. For the deterministic setting, we will add a new row to Table 1 in our paper, presented in Table 2 of the [anonymous link](https://drive.google.com/file/d/1lCY1UF3isNnoujM8AGCPIdJlRqHctj-b/view?usp=sharing) (the second page).
>
> [1] Kwon et al., A fully first-order method for stochastic bilevel optimization, ICML 2023.
>
> [2] Chen et al., Near-optimal nonconvex-strongly-convex bilevel optimization with fully first-order oracles. arXiv preprint 2023.
>
> [3] Chen et al., On finding small hyper-gradients in bilevel optimization: Hardness results and improved analysis. COLT 2024.
>
> [4] Sow et al., On the convergence theory for hessian-free bilevel algorithms, NeurIPS 2022.
>
> [5] Fang et al., qNBO: quasi-Newton Meets Bilevel Optimization, ICLR 2025.
>
> [6] Arbel et al., Amortized implicit differentiation for stochastic bilevel optimization, ICLR 2022.
>
> [7] Liu et al., Averaged Method of Multipliers for BiLevel Optimization without Lower-Level Strong Convexity, ICML 2023.

---

### Official Review · Reviewer_92Dz · 2025-03-11

**Overall Recommendation:** 3

**Summary:**

This paper consider the bilevel problem $\min_x \Phi(x) = f(x, y^*(x))$ where $y^*(x) = \arg\min_y g(x, y)$ where the inner function $g$ is strongly convex w.r.t. the inner variable $y$. The paper proposes a new AID-based method where the inner variable $y$ and the linear system variable $u$ are updated by an approximate Newton step. These approximate Newton directions are computed by applying (S)GD to the quadratic functions associated with the two linear systems. Then, the paper shows that in the deterministic setting, the proposed method improves the gradient complexity by the factor $(\kappa\log(\kappa))^{-1}$ in comparison with classical AID-based methods.  The computational complexity of the stochastic variant is also provided. Numerical expriments on a synthetic problem, hyperparameter optimization and data cleaning are provided.

**Claims And Evidence:**

* Convergence rates in the deterministic and stochastic settings are provided and sound.
* NBO-GD achieves a convergence rate in $\mathcal{O}\left(\frac{\kappa^3}{K}\right)$ leading to a gradient complexity of $\mathcal{O}\left(\frac{\kappa^3}{\epsilon}\right)$ and a HVP complexity in $\mathcal{O}\left(\frac{\kappa^4}{\epsilon}\right)$ (which can be reduced to $\mathcal{O}\left(\frac{\kappa^{3.5}}{\epsilon}\right)$ by using CG instead of GD).
* NBO-SGD achieves a convergence rate $\mathcal{O}\left(\frac{\kappa^3}{\epsilon}\right)$ (by assuming batch sizes in $\Theta(K)$) leading to a gradient complexity in $\mathcal{O}(\kappa^5/\epsilon)$ for $F$, $\mathcal{O}(\kappa^9/\epsilon)$ for $G$, and HVP complexity in $\mathcal{O}(\kappa^7/\epsilon)$.

**Essential References Not Discussed:**

To my knowledge, any essential reference is missing.

**Experimental Designs Or Analyses:**

Experimental results show the practical interest of the proposed method.

**Methods And Evaluation Criteria:**

The method is numerically evaluated on a synthetic problem, hyperparameter optimization and data cleaning. This setting is classical for the numerical evaluation of bilevel optimization methods.

**Other Comments Or Suggestions:**

* **Line 39**: when the citation is part of the sentence, it should not be in parentheses (i.e. it should be `\citet` instead of `\citep`).

* **Line 131**: *"(2) Observe that v∗(x, y) and u∗(x, y) share the same Hessian inverse."* It seems that this sentence is not supposed to be here.

* **Line 150**: *"we write $u = u^k - w$"*, $u^k$ is used before being defined.

* **Line 303**: The equation goes beyond the margin.

* **Algorithms 3 and 5**: I guess by reading the proof that it is $w^{-1},k = u^k$ instead of $w^{-1},k = 0$ in the initialization of the algorithms.

* **Theorem 3.2 and 3.7**: By reading the text, it is not clear what BOX 1 and BOX 2 refer to. These things should be introduced before the theorems.

* It would be nice to have a summury of complexity results for the stochastic setting, as done in Table 1 for the deterministic setting.

* **Box 2**: The same batches $B_0$ and $B_0'$ are used in all the iterations. Why?

* **Line 1049**: *"$\nabla_2f$"* -> *"$\nabla_2g$"*.

* **Line 1142-1143**: I think there is no $\frac12$ in factor of $\lVert y^k-y^*(x^k)\rVert^2$ anymore if the inequality comes from $(a+b)^2\leq 2a^2 + 2b^2$.

* **Section E.2 and first line of section E.2**: It Theorem 3.7 and not Theorem 3.2.

* **Equation (52)**: Isn't the equality an inequality actually?

**Other Strengths And Weaknesses:**

### Strengths
* The paper is well-written
* The method is novel to my knowledge.
* Potential improvements with variance reduction and momentum are discussed.

### Weaknesses

* The stochastic result assumes large batch sizes which scale linearly with the number of iterations. This setting does not match the practice.
* The gain in the stochastic setting is limited while this setting being important in practice.

**Questions For Authors:**

* As it is common to warm start the subsolvers in bilevel optimization [1, 2], I wonder why Algorithm 3 and 5 not set $v^{-1,k} = y^k$ instead of $v^{-1,k} = 0$.

[1] Ji, K., Yang, J., and Liang, Y. *Bilevel optimization: Convergence analysis and enhanced design*. ICML 2021
[2] Arbel M. and Mairal J. *Amortized implicit differentiation for stochastic bilevel optimization*. ICLR 2022

**Relation To Broader Scientific Literature:**

The paper proposes an adaptation of the single-loop algorithmic framework introduced in [1], where the directions for solving the inner linear system are approximated using Newton steps. It is worth noting that [2] and [3] explore the use of quasi-Newton steps to approximate the inner solution. Furthermore, [3] utilizes the quasi-Newton directions employed in solving the inner problem to enhance the resolution of the linear system. However, unlike this paper, [2] and [3] do not provide non-asymptotic results.

[1] Dagréou, M., Ablin, P., Vaiter, S., and Moreau, T. *A framework for bilevel optimization that enables stochastic and global variance reduction algorithms*. NeurIPS 2022.

[2] Pedregosa, F. *Hyperparameter optimization with approximate gradient*. ICML 2016.

[3] Ramzi, Z., Mannel, F., Bai, S., Starck, J.-L., Ciuciu, P., and Moreau, T. *Shine: Sharing the inverse estimate from the forward pass for bi-level optimization and implicit models*. ICLR, 2022.

**Theoretical Claims:**

I checked the proof for the deterministic setting and the result are sound apart from some inconsequential typos (see **Other Comments Or Suggestions** section for detail).

---

> ### Author Rebuttal · Authors · 2025-04-01
>
> Thank you for your thorough review and valuable feedback. We will address each point in detail below. The figures mentioned below are in this anonymous link: https://drive.google.com/file/d/1v6ftNYExUb_ClkgoS7b9wU2Q3nNY1IsP/view?usp=sharing
>
> ## Other Strengths And Weaknesses:
>
> * **Concern about batch size:** The large batch size we choose helps achieve better sample complexity. If the batch size is set to $O(1)$, we can obtain a sample complexity of $O(\kappa^{16} \epsilon^{-2})$, which is worse than $O(\kappa^9 \epsilon^{-2})$ in Theorem 3.7. Since SOBA is a special case of our NBO framework (as noted in Remark 2.1 (i) of our work), their analysis also applies to our NSBO-SGD when $T=0$. Moreover, a large batch size related to $ \epsilon $ or $ K $ is commonly used in the stochastic bilevel optimization literature (see, e.g., Ji et al., 2021; Arbel & Mairal, 2022).
>
> * **Our improvements in the stochastic setting and in practice:** In theory, as stated after Theorem 3.7, our NSBO-SGD improves upon the SOTA result of AmIGO by a factor of $ \log \kappa $ in the stochastic setting, where AmIGO also employs a large batch size.
>
>   In practice, our algorithms show significant advantages in experiments, even when using small batch sizes (64 or 256). Additionally, we add experiments on meta-learning using the Omniglot and miniImageNet (about 3GB) datasets, with CNN4 networks. We compared our NBO with the SOTA algorithms PZOBO in [1] and qNBO in [2]. The results, presented in Fig. 1 and 2 of the [anonymous link](https://drive.google.com/file/d/1v6ftNYExUb_ClkgoS7b9wU2Q3nNY1IsP/view?usp=sharing), demonstrate that NBO consistently outperforms the other methods on both datasets.
>
>   [1] Sow et al., On the convergence theory for hessian-free bilevel algorithms, NeurIPS 2022.
>
>   [2] Fang et al., qNBO: quasi-Newton Meets Bilevel Optimization, ICLR 2025.
>
> ## Other Comments Or Suggestions:
>
> Thank you for your careful reading. We will address each point as follows:
>
> * **Line 39:** We will revise it.
> * **Line 131:** This sentence represents the second aspect of our motivation. Although we have not fully utilized this point, we still believe it is important.
> * **Line 150:** Here we use warm-start for $u$ and will make revisions to clarify this.
> * **Line 303:** We will revise it.
> * **Algorithm 3 and 5:** The initialization point $w^{-1,k}=0$ is correct. With this initialization, we obtain $w^{0,k}=\gamma_k d_u^k$ (where $d_u^k$ is defined in Line 150), ensuring that the second equality in (44) holds.
> * **Theorem 3.2 and 3.7:** BOX 1 and 2 represent the initialization strategies of Algorithm 2 and 4, respectively. We will add an introduction before these theorems.
> * **Summary:** Depending on the available space in the final manuscript, we will consider adding a summary table for the stochastic setting.
> * **Box 2:** Thank you for pointing out the typo. $B_0$ and $B_0^{'}$ should be replaced by $B_{0,n}$ and $B^{'}_{0,q}$ respectively, while maintaining the same batch size.
> * **Line 1049:** We will revise it.
> * **Line 1142-1143:** The constant $\frac{1}{2}$ arises from $2 \times (\frac{1}{2})^2$.
> * **Section E.2 and the first line of section E.2:** We will revise them.
> * **Equation (52):** We will revise it.
>
> ## Questions For Authors:
>
> * **Concern about warm-start:** The reason we use warm-start for $u$ but not for $v$ is that $v$ directly affects $||y^{k+1}-y^{k}||$, while $u$ does not. Indeed, in the $k$-th iteration of NBO, $v^k$ serves as an inexact approximation of the Newton direction $v^*(x^k,y^k):=[\nabla_{22}^2g(x^k,y^k)]^{-1} \nabla_2 g(x^k,y^k)$, whereas $y^{k}$ approximates $y^*(x^{k})$. We update $y^{k+1}= y^k - v^k$ using a single inexact Newton step. If we further compute $v^{k+1} = v^{k} - \gamma_k d_v^{k} $ (**using one-step gradient descent with warm-start**), according to the Lyapunov function argument, we estimate $||v^{k+1} - v^*(x^{k+1},y^{k+1})|| \leq (1-\gamma_{k} \mu)||v^{k} - v^*(x^{k},y^{k})|| + ||v^*(x^{k+1},y^{k+1})-v^*(x^{k},y^{k})||. (1) $ Similar to $y^*(x)$ and $u^*(x)$, $v^*(x,y)$ is Lipschitz continuous and we can get $||v^*(x^{k+1},y^{k+1})-v^*(x^{k},y^{k})||\leq \frac{2L_{g,1}}{\mu}(||x^{k+1} - x^{k}||+||y^{k+1} - y^{k}||).          (2)$
>
>   Note that $||y^{k+1} - y^{k}||\leq ||v^k-v^*(x^{k},y^{k})||+||v^*(x^{k},y^{k})||\leq  ||v^k-v^*(x^{k},y^{k})||+\frac{L_{g,1}}{\mu}||y^{k}-y^*(x^k)||. (3)$ Substituting (2) and (3) into (1), we obtain $||v^{k+1} - v^*(x^{k+1},y^{k+1})|| \leq (1-\gamma_{k} \mu+\frac{2L_{g,1}}{\mu})||v^{k} - v^*(x^{k},y^{k})|| + others. $
>
>   Since $1-\gamma_{k} \mu+\frac{2L_{g,1}}{\mu}>1$ when $\gamma_{k}\leq 1/L_{g,1}$, the Lyapunov function argument fails to hold, indicating that the above warm-start strategy for $v$ is problematic. If multi-step gradient descent is performed for $v$, we choose to initialize at 0 because, in this setting, when $T=0$, NBO reduces to the single-loop algorithm framework proposed by (Dagréou et al., 2022) .

---

### Official Review · Reviewer_Y7jC · 2025-03-12

**Overall Recommendation:** 4

**Summary:**

This paper proposes a novel method for bilevel optimization, focusing on improving hypergradient estimation by incorporating curvature information. The key contributions include: (1) New Algorithmic Framework: An enhanced algorithm using an inexact Newton method, with improved computational complexity and convergence guarantees in both deterministic and stochastic settings. (2) Empirical Results: Numerical experiments demonstrate significant performance benefits compared to existing gradient-based methods.

"## update after rebuttal"
Through the rebuttal, I believe this work presents a general and efficient gradient-based framework for bilevel optimization, which can be extended to the non-strongly convex setting using techniques such as the BAMM method. The motivation and approach are highly interesting and are supported by strong theoretical guarantees. Therefore, I recommend acceptance.

**Claims And Evidence:**

Yes, the paper provides both theoretical and empirical evidence to support the claims made.

**Essential References Not Discussed:**

I do not find any missing references.  As far as I know, in terms of complexity, this paper compares its results with the state-of-the-art works.

**Experimental Designs Or Analyses:**

I have checked the experimental designs and analyses, including the models, datasets, and step size selection. Specifically, the step sizes are tuned via grid search, which is fair.

**Methods And Evaluation Criteria:**

Yes. In the theoretical part, the authors use $\| \nabla \Phi(x) \|$ as the stationary measure, which is reasonable for bilevel problems with a strongly convex lower-level problem. In the experimental part, the benchmark used by the authors are also reasonable, various types of bilevel algorithms are involved in the benchmark they used.

**Other Comments Or Suggestions:**

There is a citation typo in Table 1. The reference for "No-loop AID" should be (Ji et al., 2022).

**Other Strengths And Weaknesses:**

Strengths:

(1)The authors propose a novel method for estimating the hypergradient, which marks a significant departure from classical gradient-based methods. This method leverages the unique structure of the hypergradient, where a Hessian is shared when using inexact Newton method. This idea is interesting and I think it is a breakthrough.

(2)Particularly, the authors provide theoretical results that the proposed algorithm achieves lower computational complexity compared to existing methods, thereby highlighting the advantages of using curvature information.

Weaknesses:

The authors only consider the lower-level strongly convex case. (Actually, I know that the non-strongly convex case is very challenging.)

**Questions For Authors:**

Please discuss the possibility of extending the algorithm proposed in this paper to bilevel optimization problems where the lower-level problem is non-strongly convex.

**Relation To Broader Scientific Literature:**

The hypergradient method is a commonly employed approach for solving bilevel optimization problems in the literature. However, most of previous works utilized first-order methods to estimate the hypergradient. The key contribution of this paper lies in its use of the curvature information, which is theoretically proven to be more efficient by the authors. This advancement represents an improvement over existing gradient-based methods.

**Theoretical Claims:**

I have checked the proofs of Theorems 3.2 and 3.7 and did not find any errors.

---

> ### Author Rebuttal · Authors · 2025-04-01
>
> Thank you for your valuable comments and suggestions. We will address each point in detail below. The referenced figures are compiled in a single-page PDF (containing only figures) available at the anonymous link:
> https://drive.google.com/file/d/15xqtvUMRk7Ah7Gi5hnvBQWyuZ6W15zX8/view?usp=sharing
>
> ## Other Strengths And Weaknesses & Questions For Authors:
> * **Extension to lower-level non-strongly convex problems:** The NBO framework is primarily designed for bilevel optimization problems with a strongly convex structure. For non-strongly convex problems, existing methods typically involve reformulating the original problem to incorporate a strongly convex structure. Once this structure is established, our NBO framework can be applied. For instance, in BAMM method [1], when $g$ is merely convex, an aggregation function $\phi_{\mu} = \mu f + (1 - \mu) g$ is defined, which is strongly convex when $f$ is strongly convex, then an approximated hypergradient $d_x^k$ can be computed by replacing $g$ with $\phi_{\mu}$. We compared the BAMM method and BAMM+NBO (i.e., using NBO to compute $d_x^k$ in BAMM) on the toy example (13) in [1]. The result, presented in Fig. 1 of the [anonymous link](https://drive.google.com/file/d/15xqtvUMRk7Ah7Gi5hnvBQWyuZ6W15zX8/view?usp=sharing), shows that BAMM+NBO significantly outperforms BAMM, highlighting the effectiveness of the NBO framework. Due to space constraints, we primarily focused on strongly convex lower-level problems in this work, which limited our ability to fully showcase the versatility of NBO. We appreciate your valuable suggestions and will include additional discussion in the revised manuscript to further demonstrate the versatility of NBO.
>
> ## Other Comments Or Suggestions:
> Thank you for pointing this out. We will revise this citation typo.
>
> [1] Liu et al., Averaged Method of Multipliers for BiLevel Optimization without Lower-Level Strong Convexity, ICML 2023.

---

### Official Review · Reviewer_e2rS · 2025-03-14

**Overall Recommendation:** 3

**Summary:**

This paper introduces a Newton-based approach to efficiently compute hypergradients in bilevel optimization. Instead of directly inverting the Hessian, which is costly, the method approximates Hessian-inverse-vector products (HVPs) to improve computational efficiency. The proposed Newton-based bilevel optimizer (NBO) works for both deterministic and stochastic settings and provides theoretical convergence guarantees.
The main idea is to use curvature information from the inner problem to improve hypergradient estimation while keeping computation manageable. Compared to existing first-order methods, the approach reduces complexity and converges faster. Experiments on meta-learning and hyperparameter optimization show that NBO requires fewer iterations to reach comparable or better solutions.

**Claims And Evidence:**

The paper’s claims are mostly well supported by empirical evidence and theory.

**Essential References Not Discussed:**

It does not benchmark against full second-order methods (e.g., ones that directly compute Hessians rather than approximations).
It does not compare against Bregman-based methods.

**Experimental Designs Or Analyses:**

I suggest to compare against other curvature-aware algorithms.

**Methods And Evaluation Criteria:**

Consider Bregman-Based Curvature-Aware Methods:
- Would Bregman-based methods achieve similar improvements?
- Could a hybrid approach combining Newton and Bregman techniques be better?

See, e.g.,
- Enhanced Bilevel Optimization via Bregman Distance by Feihu Huang, Junyi Li, Shangqian Gao, Heng Huang
- Online Nonconvex Bilevel Optimization with Bregman Divergences by Jason Bohne, David Rosenberg, Gary Kazantsev, Pawel Polak

**Other Comments Or Suggestions:**

No

**Other Strengths And Weaknesses:**

The paper is well-structured with strong theoretical backing, but it could be improve by discussing the scalability of the method to large dimensional problems and how well it performs empirically when the strong-convexity assumption is violated.

**Questions For Authors:**

No

**Relation To Broader Scientific Literature:**

Lack of comparison against second order methods.

**Theoretical Claims:**

I did not review the proofs.

---

> ### Author Rebuttal · Authors · 2025-04-01
>
> Thank you for your valuable comments and suggestions. We will address each point in detail below. The referenced figures are compiled in a single-page PDF (containing only figures) available at the anonymous link: https://drive.google.com/file/d/1ZEzn2mKcwrPzlBeziFpC1mKGWDeyq6-C/view?usp=sharing
>
> ## Methods And Evaluation Criteria:
>
> * **Combine with Bregman-based methods:**
>   * The improvements of our NBO framework and Bregman-based methods stem from two different perspectives: the NBO framework improves hypergradient approximation (related to updates of the lower-level variable $y$), while Bregman-based methods use mirror descent to update the upper-level variable $x$.
>   * Thank you for your constructive suggestion. The idea of combining our NBO framework with Bregman-based methods is both interesting and promising. To explore this, we conducted a numerical experiment where we used the NBO framework to approximate the hypergradient, followed by updating $x$ using the mirror descent method. We then compared SBiO-BreD in [1] and SBiO-BreD+NSBO (i.e., using our proposed framework to compute $w\_t$ in SBiO-BreD) in the context of data hyper-cleaning. The result, presented in Fig. 1 of the [anonymous link](https://drive.google.com/file/d/1ZEzn2mKcwrPzlBeziFpC1mKGWDeyq6-C/view?usp=sharing), demonstrates that SBiO-BreD+NSBO significantly outperforms SBiO-BreD, highlighting the effectiveness of the NBO framework.
>
> ## Experimental Designs Or Analyses & Relation To Broader Scientific Literature:
>
> * **Comparison against other curvature-aware algorithms:** We would like to clarify that we have compared our method with SHINE in Experiments (Section 4.2 and 4.3). SHINE is a popular curvature-aware algorithm that employs the quasi-Newton method to solve the lower-level problem.
>
> ## Essential References Not Discussed:
>
> * **Comparison against exact Hessian methods:** Directly computing the Hessian and its inverse is both time-consuming and memory-consuming. In our setting, using the Hessian-vector product is a more practical choice, as it can be efficiently
>   computed and stored with modern automatic differentiation frameworks. To demonstrate this, we conducted a numerical experiment comparing NBO with the exact Hessian inverse (implemented using jax.hessian and jnp.linalg.inv) and NBO-GD for hyperparameter optimization on synthetic data. The result in Fig. 2 of the [anonymous link](https://drive.google.com/file/d/1ZEzn2mKcwrPzlBeziFpC1mKGWDeyq6-C/view?usp=sharing) shows that NBO with the exact Hessian inverse is significantly slower.
> * About comparison against Bregman-based methods, please see the response to "Methods And Evaluation Criteria".
>
> ## Other Strengths And Weaknesses:
>
> * **Scalability to large dimensional problems:** Thanks for the suggestion. As shown in Fig. 3 of the [anonymous link](https://drive.google.com/file/d/1ZEzn2mKcwrPzlBeziFpC1mKGWDeyq6-C/view?usp=sharing), we evaluated the scalability of NBO by testing hyperparameter optimization on synthetic data while progressively increasing the problem dimension.
> * **Extension to non-strongly convex problems:** The NBO framework is primarily designed for bilevel optimization problems with a strongly convex structure. For non-strongly convex problems, existing methods typically involve reformulating the original problem to incorporate a strongly convex structure. Once this structure is established, our NBO framework can be applied. For instance, we compared the BAMM method [2] and BAMM+NBO (i.e., using NBO to compute $d_x^k$ in BAMM). The result, presented in Fig. 4 of the [anonymous link](https://drive.google.com/file/d/1ZEzn2mKcwrPzlBeziFpC1mKGWDeyq6-C/view?usp=sharing), shows that BAMM+NBO significantly outperforms BAMM, highlighting the effectiveness of the NBO framework. Due to space constraints, we primarily focused on strongly convex lower-level problems in this work, which limited our ability to fully showcase the versatility of NBO. We appreciate your valuable suggestions and will include additional discussion in the revised manuscript to further demonstrate the versatility of NBO.
>
> [1] Huang et al., Enhanced bilevel optimization via bregman distance, NeurIPS 2022.
>
> [2] Liu et al., Averaged Method of Multipliers for BiLevel Optimization without Lower-Level Strong Convexity, ICML 2023.

---

### Decision · Program_Chairs · 2025-05-01

**Decision:**

Accept (poster)

**Comment:**

This paper proposes a Newton-based algorithm (NBO) for bilevel optimization, with both deterministic and stochastic variants. The authors provide convergence rate guarantees for both settings. The experimental evaluations are sufficiently comprehensive to support the main claims.

All reviewers recommend acceptance after the rebuttal. I concur with their assessment and recommend acceptance. Please incorporate necessary changes requested by the reviewers in the final version, such as addressing the scalability of the method to high-dimensional problems, discussing its performance when the strong convexity assumption is violated, and including additional comparisons to other methods mentioned in the rebuttal.